# Exon-intron boundary inhibits m⁶A deposition, enabling m⁶A distribution hallmark, longer mRNA half-life and flexible protein coding

Zhiyuan Luo[1], Qilian Ma[2], Shan Sun[2], Ningning Li[2], Hongfeng Wang ®[2], Zheng Ying ®[2] ✉ & Shengdong Ke ®[1] ✉

Regional bias of $N^6$-methyladenosine (m⁶A) mRNA modification avoiding splice site region, calls for an open hypothesis whether exon-intron boundary could affect m⁶A deposition. By deep learning modeling, we find that exon-intron boundary represses a proportion (12% to 34%) of m⁶A deposition at adjacent exons (~100 nt to splice site). Experiments validate that m⁶A signal increases once the host gene does not undergo pre-mRNA splicing to produce the same mRNA. Inhibited m⁶A sites have higher m⁶A enhancers and lower m⁶A silencers locally and show high heterogeneity at different exons genome-widely, with only a small proportion (12% to 15%) of exons showing strong inhibition, enabling more stable mRNAs and flexible protein coding. m⁶A is majorly responsible for why mRNAs with more exons be more stable. Exon junction complex (EJC) only partially contributes to this exon-intron boundary m⁶A inhibition in some short internal exons, highlighting additional factors yet to be identified.

As the most abundant mRNA internal modification, the $N^6$-methyladenosine (m⁶A) is involved in various biological processes including cell differentiation, brain development, tumorigenesis[1–6], and could affect multiple aspects of RNA metabolism, including transcription, splicing, translation, and degradation[7,8], with a major function in promoting mRNA decay[9–12]. The m⁶A is deposited to nascent pre-mRNA co-transcriptionally[11], primarily by the methyltransferase complex (MTC) comprising the catalytic core METTL3-METTL14 heterodimer and other factors[13–19]. m⁶A is installed at a motif consensus of RRACH (R = A or G, H = A, C, or U) as a stringent motif or RAC as a more inclusive motif[20–22]. Despite the wide prevalence of m⁶A consensus in mRNA, only a very small fraction is methylated[11,20]. At the global level, m⁶As reside preferentially in last exons, as well as in long internal exons[11,20]. Furthermore, m⁶As in internal exons appear to avoid the nearby exonic region close to splice sites[11]. Our previous work has revealed that the m⁶A site-specific methylation was primarily determined by the flanking nucleotide sequences, and the local functional *cis*-elements mainly resided within the 50 nt downstream of the site[23]. The underlying mechanism beyond the identification of local *cis*-regulatory elements of m⁶A site-specificity is still largely unknown.

As with m⁶A deposition, pre-mRNA splicing is also coupled with transcriptional events, allowing for potential functional crosstalk during transcription. Though several studies suggested that m⁶A could regulate alternative splicing[21,24–27], a careful bioinformatics analysis showed that loss of METTL3 in mouse embryonic stem cells had a minimal effect on pre-mRNA splicing[11]. Conversely, whether pre-mRNA splicing could affect m⁶A deposition is an open question. Most m⁶A deposition occur in the region moving away from last exon start and appears to avoid the adjacent region close to splice sites in internal exons[11,20]. These m⁶A regional distribution biases suggest that exon-intron boundary could potentially play an inhibitory role for the m⁶A deposition at the nearby region close to splice sites.

[1]The Jackson Laboratory, Bar Harbor, ME 04609, USA. [2]Jiangsu Key Laboratory of Neuropsychiatric Diseases and College of Pharmaceutical Sciences, Soochow University, Suzhou, Jiangsu 215123, China. ✉e-mail: zheng.ying@suda.edu.cn; kelab018@gmail.com

Previously we have established the iM6A deep learning model which models m⁶A site specificity with high accuracy (AUROC = 0.99) by using the primary nucleotide sequence flanking the m⁶A site[23]. This work demonstrated that the site specificity of m⁶A modification was encoded primarily by the flanking nucleotide sequence at the *cis*-level. Though the deep learning model itself is hard to be understood directly (i.e., a "black box"), we could probe for the underlying biological insights by creative in silico mutation of natural genomic regions to test our hypotheses. Then if the followed wet experiments validate randomly selected simulations, this contributes to verifying the model and the biological hypotheses it is designed to investigate. As an initial study, we performed the in silico saturation mutagenesis on the local sequences surrounding the m⁶A site and discovered that the downstream 50 nt region of the m⁶A site was highly enriched with the *cis*-elements governing m⁶A deposition[23]. Independent experimental validation supported this finding. The in silico deep learning modeling approach has proved to be an effective way to investigate the *cis*-regulatory mechanisms that determines m⁶A deposition, and offers a high-throughput and fast-paced low-cost discovery mechanism relative to exclusively experimental studies which could be cost-prohibitive[23].

In this study, we implemented iM6A deep learning modeling to investigate *cis*-regulatory mechanisms for m⁶A site specificity beyond the local *cis*-regulatory elements. By the in silico mutational modeling at gene intron deletions, we discovered that exon-intron boundary inhibits a proportion of m⁶A deposition at nearby exons. These inhibited m⁶A sites tended to have a good local *cis*-element environment with more m⁶A enhancers and fewer m⁶A silencers, compared to the m⁶A sites that were not inhibited. These modeling findings were supported by the experimental validation, as will be shown below. The m⁶A deposition inhibition by exon-intron boundary exhibited a high heterogeneity at genomic level, with a small proportion of exons exhibiting strong inhibition. By this m⁶A deposition inhibition mechanism by exon-intron boundary, multi-exon mRNA will have longer half-life given the same primary nucleotide sequence and m⁶A is a major contributor to mRNAs with more exons tend to be more stable; Also, this mechanism enables mRNA to encode protein sequence flexibly with less concern of creating too many m⁶A sites to compromise its mRNA stability.

## Results

### Deep learning modeling revealed that exon-intron boundary inhibits m⁶A deposition at last exon and second-to-last exon

As we previously found that m⁶A appeared to avoid the nearby region close to splice sites while being mostly enriched in the region moving away from last exon starts[11,20], we speculated that exon-intron boundary might inhibit m⁶A deposition at exons. We modeled this with an in silico mutational experiment by deleting the last intron sequences from each gene to generate the non-last intron genes as the input for iM6A (Fig. 1a) (i.e., pre-mRNA would not undergo pre-mRNA splicing of last intron to generate mRNA). We unexpectedly found that the m⁶A density increased around last exon start (Fig. 1a for mouse, and Supplementary Fig. 1a for human).

A more detailed examination down to individual RAC sites in this region revealed that (1) a proportion of RAC sites (~12%) in last exons had an increase in m⁶A deposition (Fig. 1b for mouse, and Supplementary Fig. 1b for human). Since the m⁶A deposition of these sites were repressed by the exon-intron boundary of last intron, we define them as the repressed m⁶A sites or latent m⁶A sites; (2) most of those sites were enriched within the -100 nt region to last exon start (Fig. 1b for mouse, and Supplementary Fig. 1b for human). Next, we split last exons into three groups based on its length (<= 200, 200 400, and >= 400 nt), and these latent sites were enriched in the -100 nt region to last exon start for all three groups (Supplementary Fig. 2), demonstrating that m⁶A deposition inhibition by exon-intron boundary

occurs near the splicing sites for both short and long exons. In our previous publication of the iM6A deep learning modeling[23], we implemented a high-throughput in silico saturated point mutations around m⁶A sites and discovered that the local *cis*-elements that regulating m⁶A site-specificity are highly enriched in the downstream 50 nt region. Furthermore, from such an over one million point-mutation modeling events, we calculated out the quantitative contributions of m⁶A site-specificity by each of the total 1024 pentamers using a linear regression model: m⁶A enhancers are top ranked 5mers (i.e. enhancing m⁶A deposition) while m⁶A silencers are bottom ranked 5mers (i.e. silencing m⁶A deposition).

We further investigated the distribution of m⁶A enhancers and m⁶A silencers in the local region flanking the RAC sites upon last intron deletion. In comparison to the majority RAC sites without m⁶A deposition change, the RAC sites with increased m⁶A deposition contained more m⁶A enhancers in the downstream 50 nt region (Fig. 1c for mouse, and Supplementary Fig. 1c for human) while hosting less m⁶A silencers in the same region (Fig. 1d for mouse, and Supplementary Fig. 1d for human). This data showed that those latent m⁶A sites (ΔProbability > 0.1) in last exons had a favorable local *cis*-element composition for m⁶A deposition but was repressed by exon-intron boundary. Evolution conservation analysis showed that these repressed m⁶A sites were more conserved in comparison to the RAC sites that were not subject to this exon-intron boundary inhibition (Fig. 1e, f for mouse, and Supplementary Fig. 1e, f for human), supporting their functional importance.

Besides repressing the m⁶A deposition in last exons, exon-intron boundary might also inhibit the m⁶A deposition in the second-to-last exons. We examined the m⁶A change situation in second-to-last exon to demonstrate that the inhibitory effect of exon-intron boundary exists locally in the 100 nt splice-site-adjacent exonic region of the two flanking exons. We found the increase of m⁶A deposition (due to the deletion of last intron) occurred only locally in the second-to-last exon as well as last exon, without affecting other upstream exons (Fig. 1g for mouse, and Supplementary Fig. 1g for human). Next, we plotted the detailed m⁶A methylation changes for all the RAC sites in the second-to-last exons. Upon the last intron deletion, ~22% RAC sites had increased m⁶A probability (Fig. 1h for mouse, and Supplementary Fig. 1h for human), and most of those latent sites were also enriched in the -100 nt region close to the end of second-to-last exons (Fig. 1h for mouse, and Supplementary Fig. 1h for human). Similarly, those latent sites were enriched in the -100 nt region close to second-to-last exon ends for both short and long exons (Supplementary Fig. 3). Also, the m⁶A enhancers enriched and m⁶A silencers avoided in the 50 nt downstream region of these latent m⁶A sites respectively (Fig. 1i, j for mouse, and Supplementary Fig. 1i, j for human). These data demonstrated that exon-intron boundary inhibits the local m⁶A deposition at its two adjacent exons while not affecting other upstream exons (Fig. 1g for mouse, and Supplementary Fig. 1g for human). In addition, these repressed m⁶A sites were also more conserved in comparison to the RAC sites that were not subject to this intron inhibition suggesting their functional importance (Fig. 1k, l for mouse, and Supplementary Fig. 1k, l for human).

### Deep learning modeling revealed that exon-intron boundary inhibits m⁶A deposition at internal exons

It is possible that exon-intron boundary also inhibits m⁶A deposition in internal exon. To test this hypothesis, we performed a new round of m⁶A deposition in silico modeling by deleting all introns from the gene (i.e. pre-mRNA would not undergo pre-mRNA splicing to generate mRNA), and found that the m⁶A level at internal exons also increased remarkably upon intron deletion (Fig. 2a–c for mouse, and Supplementary Fig. 4a–c for human). Overall ~34% RAC sites in internal exons showed higher m⁶A probability (Fig. 2b, c for mouse, and Supplementary Fig. 4b, c for human), and those latent m⁶A sites also mostly

resided in the ~100 nt region to the two ends of internal exons (Fig. 2b, c for mouse, and Supplementary Fig. 4b, c for human). Given that most internal exons in vertebrate are short (average size <150 nt)[28], detail examinations down to different exon length (<= 200, 200 – 400, and >= 400 nt) revealed that the m[6]A deposition inhibited by exon-intron boundary specifically occurred 100 nt near the splicing sites, even in long exons (Fig. 2d–i for mouse, and Supplementary Fig. 4d–i for human). In addition, the m[6]A enhancers or silencers were enriched or avoided in the 50 nt downstream region of these repressed m[6]A sites respectively, again supporting that these repressed m[6]A sites had a

good local *cis*-elements composition for m[6]A deposition but were repressed by the nearby exon-intron boundary (Fig. 2j, k for mouse, and Supplementary Fig. 4j, k for human). Evolution conservation analysis demonstrated that these repressed m[6]A sites were more conserved in comparison to the RAC sites that were not subject to this exon-intron boundary inhibition (Fig. 2l, m for mouse, and Supplementary Fig. 4l, m for human).

To further understand the m[6]A inhibition by exon-intron boundary, we truncated either last intron (Supplementary Fig. 5a for mouse, and Supplementary Fig. 5c for human) or all introns (Supplementary

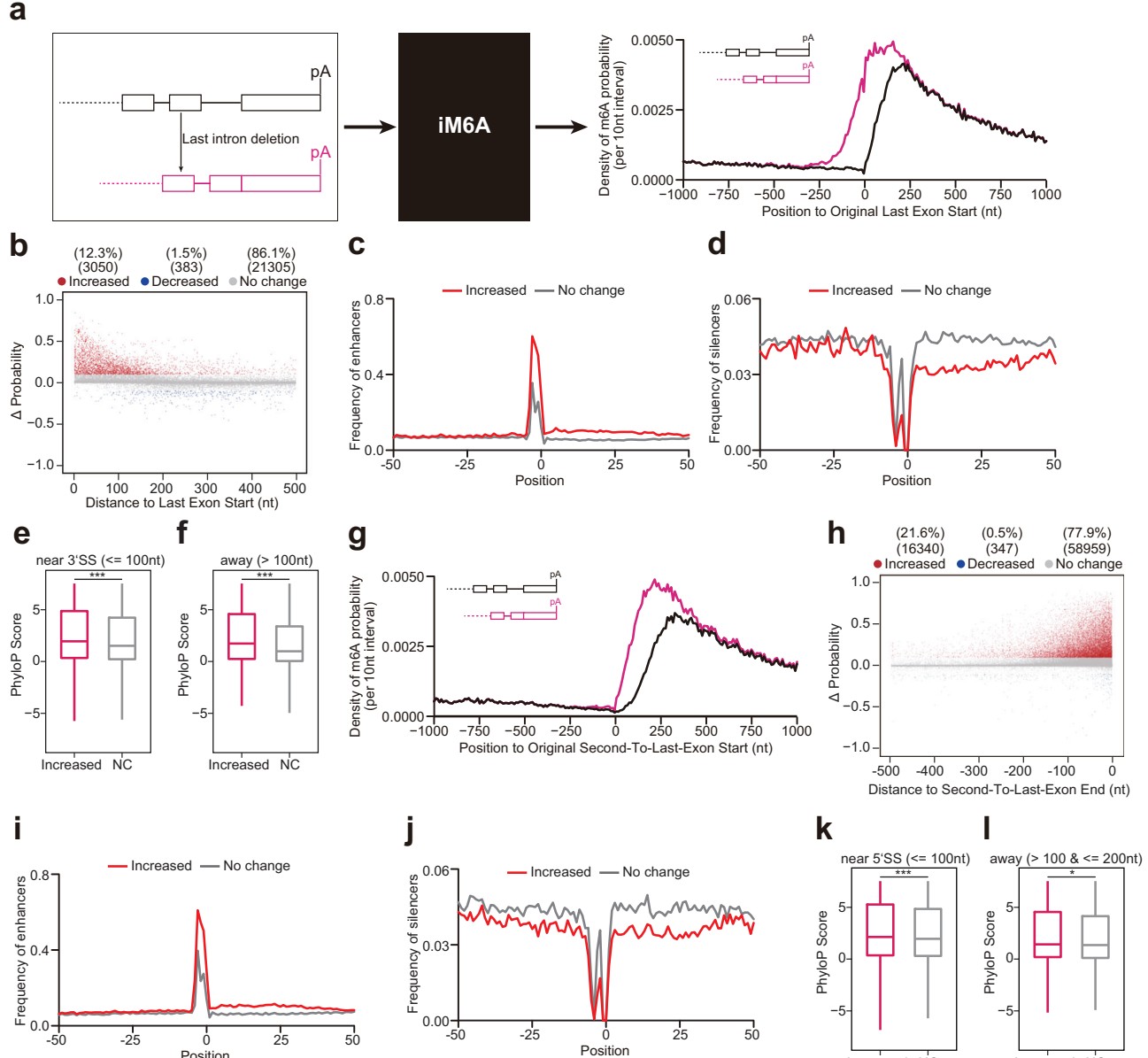

**Fig. 1 | Deep learning modeling reveals last intron deletion revives m[6]A deposition at the local adjacent exonic regions of last exon and second-to-last exon. a, g** Schematic figure of in silico modeling m[6]A deposition in pre-mRNA by iM6A. The m[6]A density of transcripts around original last exon start (**a**) or second-to-last exon start (**g**) were compared between full length (black line) and last intron deletion (pink line). **b, h** Positional plot of ΔProbability for the RAC sites located in the first 500 nt region of last exons (**b**, 2000 genes) or the last 500 nt region of original second-to-last exons (**h**, 16769 genes). Red, blue, and gray dots were those sites that had increased (> 0.1), decreased (< −0.1), or not change probability (|ΔProbability| <= 0.1) respectively by last intron deletion. **c, d, i, j** Positional plot for

the frequency of top 50 m[6]A enhancers (**c, i**), m[6]A silencers (**d, j**) in mRNA sequences around the RAC sites in last exon (**c, d**) or second-to-last exon (**i, j**). Increased sites (red line, ΔProbability > 0.1), and no change sites (gray line, | ΔProbability| <= 0.1). **e, f** Box plot of PhyloP score of latent m[6]A sites or no change sites in near 3'SS (**e**, $p < 2.22e^{-16}$) or away (**f**, $p < 2.22e^{-16}$) from 3'SS (n = 14724 or 32262 for **e**, $n = 4729$ or 110760 for **f**). **k, l** Box plot of PhyloP score of latent m[6]A sites or no change sites in near 5'SS (**k**, $p = 2.3e^{-6}$) or away (**l**, $p = 0.028$) from 5'SS ($n = 10608$ or 28074 for **k**, $n = 1117$ or 3511 for **l**). In (**e, f, k, l**) the box represents the 1st to 3rd quartile with the median marked by a horizontal line, the *P*-values were calculated by two-sided Wilcoxon test (Significance: ***$p < 0.001$, *$p < 0.05$).

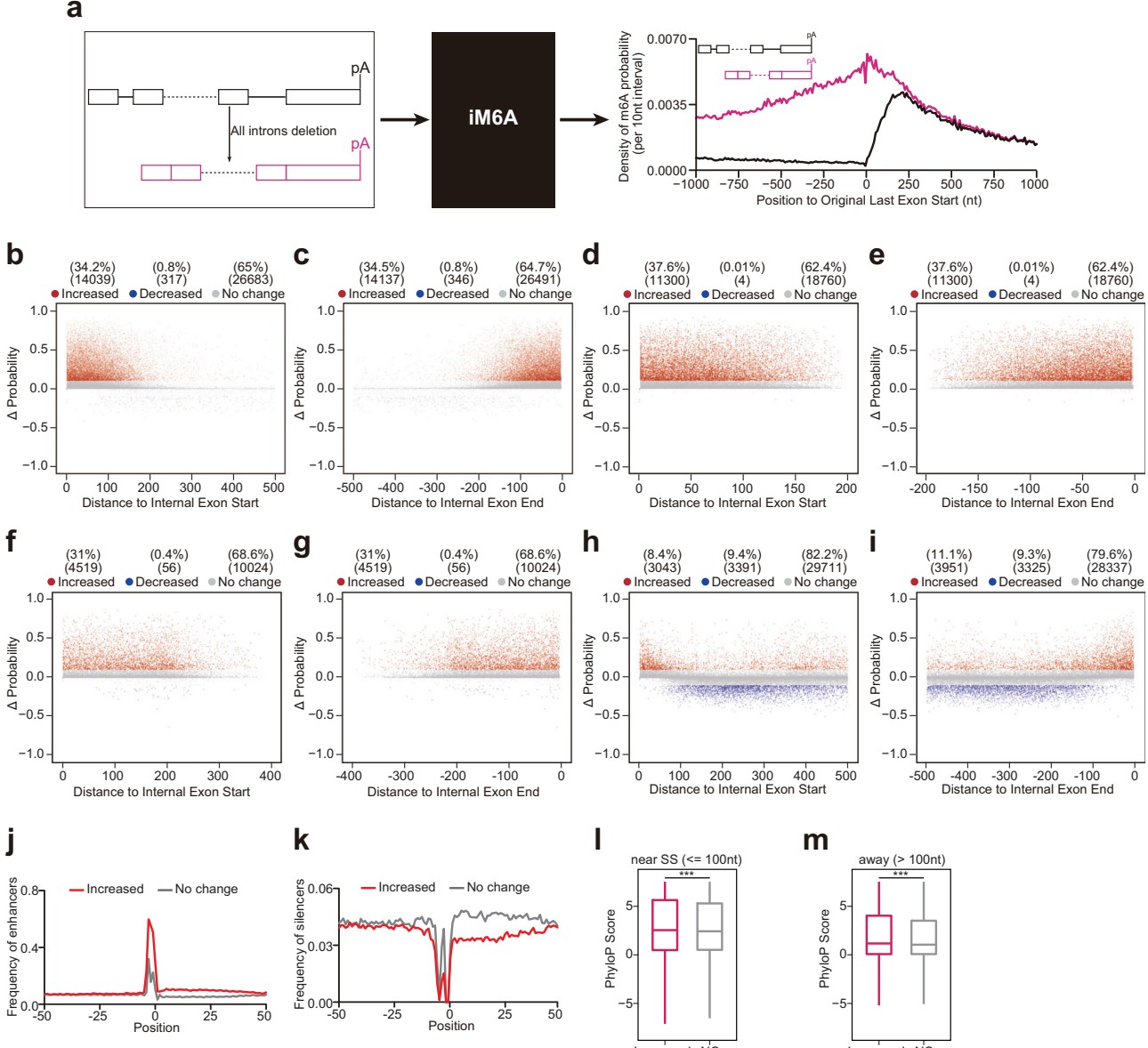

**Fig. 2 | Deep learning modeling reveals introns deletion revives m⁶A deposition at splice site adjacent exonic regions of internal exons. a** The m⁶A density of transcripts around original last exon start were compared between full length (black line) and all introns deletion (pink line). For all introns deletion, the input for iM6A is the nucleotide sequence of mRNAs (only exon sequences with no introns). **b, c** Positional plot of ΔProbability for the RAC sites located in the first (**b**, 1000 genes) or last (**c**, 1000 genes) 500 nt region of internal exons. Red, blue, and gray dots were those sites that had increased (> 0.1), decreased (< −0.1), or not change probability (|ΔProbability| <= 0.1) respectively by introns deletion. **d–i** Positional plot of ΔProbability for the RAC sites located in the first or last 200 (**d**, e, 1000 genes), 400 (**f, g**, 1000 genes), 500 (**h, i**, 2000 genes) nucleotide region of internal

exons (exon length: <= 200 nt for (**d, e**), > 200 nt & < 400 nt for (**f, g**), >= 400 nt for (**h, i**)). Red, blue, and gray dots were those sites that had increased (> 0.1), decreased (< −0.1), or not change probability (|ΔProbability| <= 0.1) respectively by introns deletion. **j, k** Positional plot for the frequency of top 50 m⁶A enhancers (**j**), m⁶A silencers (**k**) in mRNA sequences around the RAC sites. Increase sites (red line, ΔProbability > 0.1), and no change sites (gray line, |ΔProbability| <= 0.1). **l, m** Box plot of PhyloP score of latent m⁶A sites or no change sites in near (**l**, $p < 2.2e^{-16}$) or away (**m**, $p = 2.1e^{-5}$) from splice sites ($n = 200,565$ or $339,928$ for (**l**), $n = 8727$ or $89,545$ for (**m**)). The box represents the 1st to 3rd quartile with the median marked by a horizontal line. The P-values were calculated by two-sided Wilcoxon test (Significance: ***$p < 0.001$.).

Fig. 5b for mouse, and Supplementary Fig. 5d for human) to a maximum of 400 nucleotides by keeping the nearest 200 nucleotides at the two intron ends (original mean intron length: ~4.8 kb for mouse, and ~6 kb for human). As intronic splicing *cis*-elements are highly enriched at the 100 nt flanking intronic region of most human and mouse exons[29], these mini-introns should mostly retain their splicing capacity. Intron size reduction only altered the m⁶A density mildly (Supplementary Fig. 5a, b for mouse, and Supplementary Fig. 5c, d for human), suggesting that the deep intronic sequences only played a minor role in inhibiting m⁶A deposition at nearby exons. We further

truncated the full-length last introns to 200 nucleotides mini-introns by preserving the flanking 100 nucleotides of the two intron ends which contain highly enriched intronic splicing *cis*-elements[29] (Supplementary Fig. 6a–c). As above, the deep intronic sequence contributed little to this m⁶A deposition inhibition (Supplementary Fig. 5), and the m⁶A density at the ends of the two flanking exons had little change upon this intron length truncation (Supplementary Fig. 6a–c). In contrast, the deletion of mini-introns promoted m⁶A deposition at ~100 nt region of the two nearby exons (Supplementary Fig. 6a–c). These data support that the exon-intron boundary of the 200 nt long

mini-intron may be as potent in inhibiting m⁶A deposition at nearby exons as the exon-intron boundary of the full-length intron, enabling the minigene experimental validation below. In our previous work, we systematically characterized pentamer motifs as m⁶A enhancers and silencers and demonstrated their respective contributions to m⁶A deposition by independent experimental validations[23]. We speculated that local motifs in introns might not be in favor of m⁶A deposition. To verify it, we compared the distribution of m⁶A enhancers/silencers in the retained introns and the exonic sequences. The exonic sequences had a higher frequency of m⁶A enhancers than silencers (Supplementary Fig. 6d for mouse, and Supplementary Fig. 6e for human), and m⁶A silencers were particularly enriched in each intronic end of the retained mini-introns (i.e. splice site region, Supplementary Fig. 6d, e).

### Experimental validation of exon-intron boundary inhibition on m⁶A deposition

To experimentally validate the exon-intron boundary inhibition on m⁶A deposition, we ligated the coding sequence (CDS) of AcGFP1 in-frame to a minigene. The minigene consisted of two exons and a 200 nt intervening mini-intron (Fig. 3a). We constructed two such minigenes, *Lrp12* and *Gne*. The pre-mRNA splicing of both minigenes occurred efficiently (Fig. 3b, and Supplementary Fig. 20), experimentally confirming that the 200 nt long mini-intron retained its splicing capacity. The iM6A modeling predicted the m⁶A inhibition by exon-intron boundary in both minigenes, *Lrp12* and *Gne* (Supplementary Fig. 7a, b). Consistently, using the SELECT method to experimentally quantify m⁶A[30], we did observe the m⁶A signal increase in both minigenes when they did not undergo pre-mRNA splicing to produce the mRNA with the same nucleotide sequence (Fig. 3c, d). Altogether, eight RAC sites were predicted to increase their m⁶A level when the minigene did not undergo pre-mRNA splicing to produce the mRNA with the same nucleotide sequence (predicted m⁶A level increase > 0.1) (Supplementary Fig. 7a), and five such RAC sites were experimentally confirmed to increase their m⁶A level (highlighted in Fig. 3c, d). We experimentally quantified all 19 RAC sites both minigenes and found that they overall had an evident m⁶A signal increase (average relative m⁶A level increase = 0.264 > 0, $p = 0.029$, one sample $t$-test) (Fig. 3e), agreeing with the iM6A prediction (average predicted methylation level increase = 0.197 > 0, $p = 0.0004$, one sample $t$-test) (Supplemental Fig. 7b). These experimental data confirmed that exon-intron boundary inhibits m⁶A deposition at nearby exons (Fig. 3, and Supplementary Fig. 7). At the same time, we observed the RAC sites in individual nearby exons had distinct m⁶A deposition inhibition, some exons were strongly inhibited by exon-intron boundary, while others were not (Fig. 3c, d), suggesting heterogeneity of m⁶A deposition inhibition.

Since a major function of m⁶A is promoting mRNA decay[9–12], the mRNA produced without pre-mRNA splicing inhibition has stronger m⁶A signal, and thus should have shorter half-life ($T_{1/2}$). As expected, for both *Lrp12* and *Gne*, the mRNAs produced by constructs that didn't undergo pre-mRNA splicing had shorter $T_{1/2}$s than mRNAs produced by constructs that did undergo pre-mRNA splicing, though these two mRNAs shared identical primary nucleotide RNA sequence (Fig. 3f, g).

### A small proportion of last exons exhibit strong m⁶A deposition inhibition by exon-intron boundary

As we observed distinct m⁶A deposition inhibition by exon-intron boundary in individual flanking exons in the validation experiments (Fig. 3), we further comprehensively investigated this exon heterogeneity of m⁶A deposition inhibition at a genome-wide scale. Towards this goal, we calculated the m⁶A probability change (ΔProbability) for the RAC sites located in all last exons after the last intron deletion in the gene for each gene in this study. The first 200 nucleotides of last exons were binned into 40 interval (5 nucleotides per interval). In each interval, the RAC site with maximum probability change was selected,

and its corresponding ΔProbability was calculated as the ΔValue for the interval. Then based on the ΔValue and using the k-means clustering method, we clustered all the last exons into two groups: Cluster1 (C1) and Cluster2 (C2) (Fig. 4a for mouse, and Supplementary Fig. 8a for human). C1 exons were those highly enriched with the signal increased m⁶A sites (Fig. 4a for mouse, and Supplementary Fig. 8a for human), indicating C1 exons exhibited strong m⁶A deposition inhibition by exon-intron boundary. We found that ~30% RAC sites in C1 exons showed increased m⁶A deposition (Fig. 4b for mouse, and Supplementary Fig. 8b for human), which was threefold of that in C2 exons (Fig. 4c for mouse, and Supplementary Fig. 8c for human). Furthermore, these repressed m⁶A sites (ΔProbability > 0.1) were enriched in the ~100 nt region of the C1 exons start (Fig. 4b for mouse, and Supplementary Fig. 8b for human), and in both short and long exons (Supplementary Fig. 9). To further investigate these two distinct exon groups, we plotted their m⁶A levels before and after last intron deletion respectively. The m⁶A level at C1 exons was only mildly higher than that in C2 exons before last intron deletion in the gene (Fig. 4d–f for mouse, and Supplementary Fig. 8d–f for human). However, after last intron deletion in the gene, the m⁶A density increased sharply at C1 exons (about fivefold), but not at C2 exons (Fig. 4e–g for mouse, and Supplementary Fig. 8e–g for human). To understand the underlying *cis*-element mechanism in the C1 and C2 exons, we compared the distribution of m⁶A enhancers and silencers around these repressed m⁶A sites to that of RAC sites without m⁶A deposition change. The m⁶A enhancers were more enriched in the 50 nt downstream of the repressed m⁶A sites in C1 exons (Fig. 4h, i for mouse, and Supplementary Fig. 8h, i for human), while the silencers were more avoided this region in comparison to these sites in C2 exons (Supplementary Fig. 13a, b for mouse, and Supplementary Fig. 13c, d for human). In addition, we found the RAC sites were strongly enriched (about twofold) in the ~100 nt region of exon start in C1 exons in comparison to that in C2 exons (Fig. 4j–l for mouse, and Supplementary Fig. 8j–l for human).

We examined all the pentamer occurrence comparing C1 vs. C2. The NRACN motifs (i.e. RAC containing pentamer) were more likely to be enriched in C1 exons (Fig. 4m for mouse, and Supplementary Fig. 8m for human). In addition, m⁶A enhancers were also more enriched in C1 exons, while the m⁶A silencers were more avoided (Fig. 4n for mouse, and Supplementary Fig. 8n for human), supporting our findings that C1 exons tend to be with better local *cis*-element environment than C2 exons. We also showed the 20 most enriched or avoided motifs. The 20 most enriched motifs included many parts of the RRACH motif (Fig. 4o for mouse, and Supplementary Fig. 8o for human), and the 20 most avoided motifs contained CG dinucleotides (Fig. 4p for mouse, and Supplementary Fig. 8p for human). We also compared the exon lengths and 3'-UTR lengths between C1 and C2 last exons. Both exon length and 3'-UTR length of C1 exons were longer than C2 (Supplementary 10a, b for mouse, and Supplementary Fig. 10c, d for human). Altogether, the m⁶A deposition inhibition by exon-intron boundary in last exons demonstrated a high heterogeneity: only a small proportion (mouse: 12.3%, 2339 out of 19045; human: 14.7%, 2681 out of 18209) of last exons exhibited strong inhibition, and these last exons contained a high density of RAC and m⁶A enhancer motifs and low density of m⁶A silencer motifs in the first 100 nt region of the last exon start.

### A small proportion of internal exons exhibit strong m⁶A deposition inhibition by exon-intron boundary

We speculated that internal exons might also demonstrate a high heterogeneity for m⁶A deposition inhibition by exon-intron boundary. Accordingly, for the RAC sites located in internal exons, we calculated the m⁶A probability change (ΔProbability) after all introns were deleted in the gene, and applied the k-means method to cluster the internal exons into two groups: Cluster1 (C1) and Cluster2 (C2) (Fig. 5a

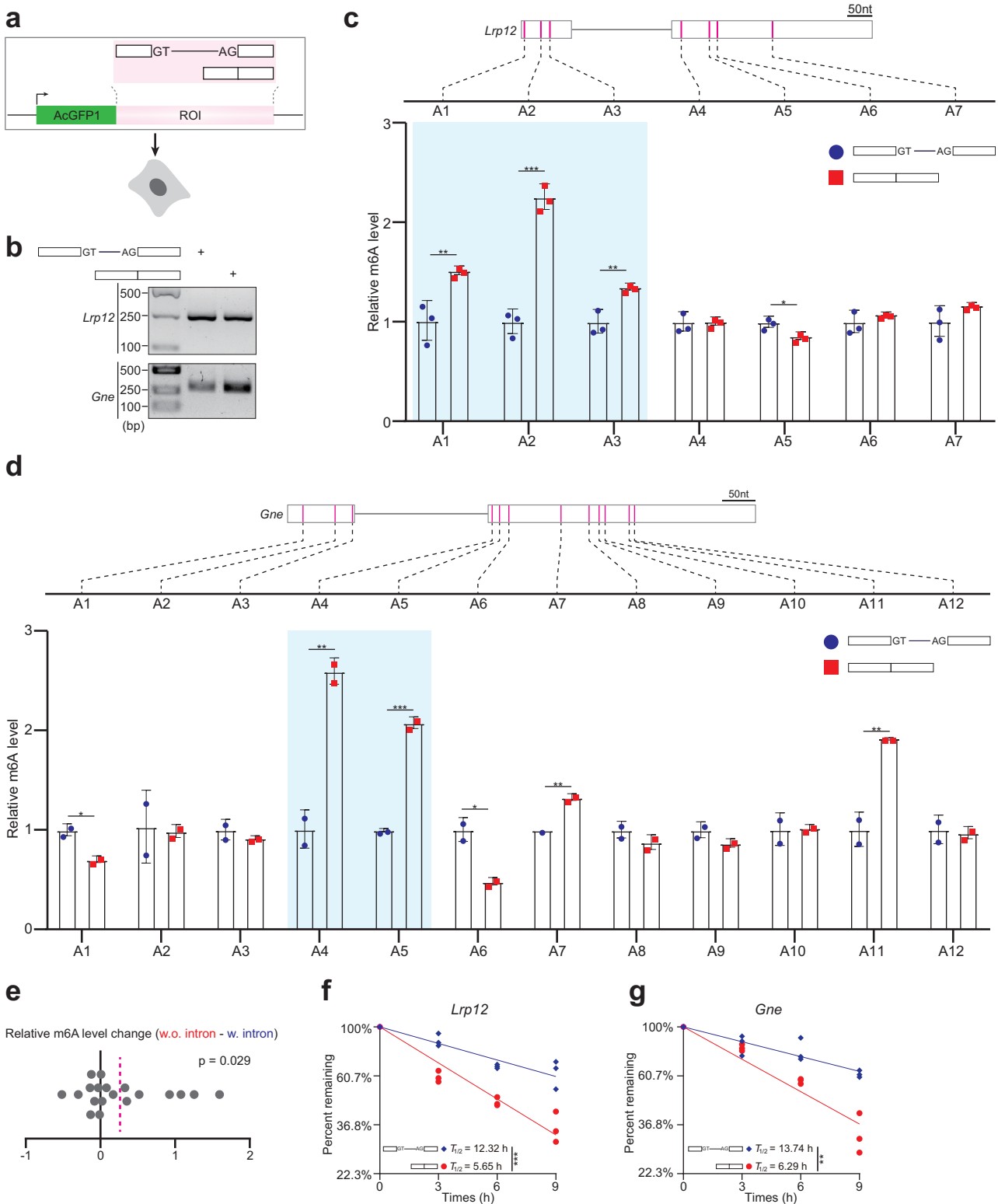

for mouse, and Supplementary Fig. 11a for human). C1 exons were highly enriched with the increased m⁶A deposition sites (Fig. 5a for mouse, and Supplementary Fig. 11a for human), exhibiting strong m⁶A deposition inhibition by pre-mRNA splicing. In total, ~70% of RAC sites in C1 exons showed increased m⁶A deposition (Fig. 5b for mouse, and Supplementary Fig. 11b for human), which was about 3-fold of that in C2 exons (Fig. 5c for mouse, and Supplementary Fig. 11c for human), and in both short and long exons (Supplementary Fig. 12). Furthermore, the repressed m⁶A sites (ΔProbability > 0.1) were enriched in the

-100 nt region of C1 exon start (Fig. 5b and Supplementary Fig. 6b). Before intron deletion in the gene, the m⁶A levels at internal exons were very low in both C1 and C2 exons (Fig. 5d–f for mouse, and Supplementary Fig. 11d–f for human). After intron deletion, the m⁶A density increased sharply at C1 exons, not at C2 exons (Fig. 5e–g for mouse, and Supplementary Fig. 11e–g for human).

Consistent with the m⁶A enhancer and silencer distribution flanking RAC sites in last exons, the m⁶A enhancers were more enriched in the 50 nt downstream of increased sites in C1 exons (Fig. 5h, i

**Fig. 3 | Experimental validation of intron repression on m⁶A deposition.**
**a** Illustration of the minigene experiment, using *Lrp12* and *Gne* as two model genes. The minigene contained two exons and a 200 nt intervening mini-intron (details in Methods), and the first exon was in-frame fused to AcGFP1. Constructs were transfected into HEK293T cells, and m⁶A signal was quantified by SELECT method. **b** The pre-mRNA splicing of mini-intron in minigene of *Lrp12* or *Gne* was validated by RT-PCR in HEK293T cells. At least three independent experiments were performed. **c, d** The bar plot of relative m⁶A level for detecting the m⁶A sites in mRNA. The constructs of minigenes were shown, and RAC sites in *Lrp12* (**c**) or *Gne* (**d**) were marked as pink lines. Data were presented as mean ± SD, the *P*-values were calculated by two-sided Student's *t*-test ($p = 0.0065$, 0.00013, 0.004, 0.011 for A1, A2, A3, A5 in *Lrp12*, $p = 0.013$, 0.005, 0.0008, 0.014, 0.003, 0.009 for A1, A4, A5, A6, A7, A11

in *Gne*. Significance: ***$p < 0.001$, **$p < 0.01$, *$p < 0.05$. $n = 3$ or 2 biological independent samples in (**c**) or (**d**)). The RAC site showed agreed increased m⁶A signal in both iM6A modeling and experimental validation was marked by blue box. **e** The dot plot of the experimental determined relative m⁶A level change for each RAC site in mRNA (19 sites in total). Relative m⁶A level change was calculated for the relative m⁶A level of each site between intron-containing and intron-deletion mRNAs. The mean value (0.264) was shown as the dotted pink line, and *P*-value was calculated by one-sample one-sided *t*-test for the increase vs. no change. **f, g** mRNA decay plotted as a function of time. The normalized levels of minigene mRNA at 0 h were set to 100%. The *y* axis represents the log value of mRNA remaining level. The *P*-values were calculated by two-sided Student's *t*-test ($p = 5.4e^{-6}$ in (**f**) for *Lrp12*, $p = 0.0026$ in (**g**) for *Gne*. Significance: ***$p < 0.001$, **$p < 0.01$).

for mouse, and Supplementary Fig. 11h, i for human), while the silencers tended to be avoided this region (Supplementary Fig. 13e, f for mouse, and Supplementary Fig. 13g, h for human). Lastly, the RAC sites were about 2 fold enriched in the -100 nt region of exon start in C1 exons comparing to that in C2 exons (Fig. 5j–l for mouse, and Supplementary Fig. 11j–l for human). Pentamer occurrence were also compared between C1 and C2. Similarly, the RAC-containing pentamers were more likely to be enriched in C1 exons (Fig. 5m for mouse, and Supplementary Fig. 11m for human). Moreover, m⁶A enhancers were more enriched in C1 exons, while m⁶A silencers were more avoided (Fig. 5n for mouse, and Supplementary Fig. 11n for human). The 20 most enriched or avoided motifs were showed: the 20 most enriched motifs included many parts of the RRACH motif (Fig. 5o for mouse, and Supplementary Fig. 11o for human), and the 20 most avoided motifs contained CG dinucleotides (Fig. 5p for mouse, and Supplementary Fig. 11p for human). m⁶A deposition inhibition by exon-intron boundary occurs at both end of internal exons. Accordingly, to be comprehensive, we clustered the internal exons into two groups based on ΔProbability at exon end region (Supplementary Fig. 14 for mouse, Supplementary Fig. 15 for human), and came to same conclusions (Supplementary Figs. 14–17). In summary, the m⁶A deposition inhibition by exon-intron boundary in internal exons also had a high heterogeneity at both exonic ends, and a small proportion of internal exons exhibited strong inhibition.

### The m⁶A deposition inhibition by exon-intron boundary allows longer mRNA half-life
Since the exon-intron boundary inhibits m⁶A deposition at the nearby exons, one would expect an anti-correlation between the m⁶A deposition efficiency and the pre-mRNA splicing events (i.e. exon number) in the host genes. Indeed, in our minigene validation (Fig. 3), we experimentally confirmed this hypothesis. To extend this finding at a genome-wide scale, we performed the scatter density plot between m⁶A/RAC ratio and the exon number in individual mRNAs, and observed a strongly negative correlation between the pre-mRNA splice events and m⁶A/RAC ratio (i.e. m⁶A deposition inhibition by exon-intron boundary) (Fig. 6a). Individual mRNAs with higher exon number had lower m⁶A deposition efficiency (Fig. 6a, and Supplementary Fig. 18a, b). Since a major function of m⁶A mRNA modification is to promote mRNA decay[9–12], mRNAs with short half-lives ($T_{1/2}$s $< 5$ h) had higher rate of m⁶A deposition, while mRNAs with longer half-lives ($T_{1/2}$s of 5–10 h or >10 h) had a progressively lower rate of m⁶A deposition (Fig. 6b). However, this negative correlation between $T_{1/2}$s and rate of m⁶A deposition vanished in mRNAs of *Mettl3* knockout mESCs (Fig. 6c), highlighting that this correlation is dependent on m⁶A. Similarly, mRNAs with short half-lives ($T_{1/2}$s $< 5$ h) had fewer exons, while mRNAs with $T_{1/2}$s of 5-10 h or > 10 h had a progressively increased exon number (Fig. 6d, and Supplementary Fig. 18c). In addition, this correlation between $T_{1/2}$s and exon numbers in individual mRNAs was also lost in *Mettl3* knockout mESCs (Fig. 6e, and Supplementary Fig. 18d). To sum up, m⁶A mRNA modification accounts majorly for the correlation that multi-exon genes have more stable mRNAs.

Having shown that m⁶A deposition efficiency is anti-correlated with pre-mRNA splicing events, it would be reasonable that mRNAs with fewer exons may have higher m⁶A levels. To test this hypothesis, we compared the m⁶A level between single-exon and multiple exon genes by matching RAC sites in mRNAs (Fig. 6) or match cDNA length (Supplementary Fig. 18). We found that single-exon genes had higher number of m⁶A sites than multiple-exon genes (Fig. 6f and Supplementary Fig. 18e). Since m⁶A negatively regulates mRNA half-life, these single-exon genes had shorter $T_{1/2}$s (Fig. 6g, and Supplementary Fig. 18f) and greater $T_{1/2}$s changes between *Mettl3* KO vs WT mESC cells (Fig. 6i and Supplementary Fig. 18h). Moreover, the difference of $T_{1/2}$s between single-exon and multiple-exon genes was lost upon global loss of m⁶A in *Mettl3* KO mESC cells (Fig. 6h and Supplementary Fig. 18g). We performed a further analysis and found that mRNAs with 2–6 exons also had higher number of m⁶A sites than mRNAs with >= 7 exons (Fig. 6j and Supplementary Fig. 18i), and mRNAs with 2-6 exons also had shorter $T_{1/2}$s (Fig. 6k and Supplementary Fig. 18j) and greater $T_{1/2}$s changes between Mettl3 KO vs WT mESC cells (Fig. 6m and Supplementary Fig. 18l). Although $T_{1/2}$s of mRNAs with 2–6 exons were shorter in *Mettl3* knockout mESCs (Fig. 6l and Supplementary Fig. 18k), the difference of $T_{1/2}$s (2–6 exons vs. >=7 exons) was much smaller than that in *Mettl3* WT mESCs.

Since we discovered that m⁶A deposition was strongly inhibited in a small proportion of exons (C1 exons), we speculated that mRNAs with C1 exons would have lower m⁶A levels than these without C1 exons. As expected, mRNAs with C1 exons had fewer number of m⁶A sites (Fig. 6n and Supplementary Fig. 18m), longer $T_{1/2}$s (Fig. 6o and Supplementary Fig. 18n) and smaller $T_{1/2}$s changes between Mettl3 KO vs WT mESC cells (Fig. 6q and Supplementary Fig. 18p). In addition, the difference of $T_{1/2}$s (C1 vs C2) was almost lost upon global loss of m⁶A in Mettl3 KO mES cells (Fig. 6p and Supplementary Fig. 18o). These data collectively demonstrate that exon-intron boundary inhibits m⁶A deposition, allowing longer mRNA half-life for mRNAs with more exons.

### The m⁶A deposition inhibition by exon-intron boundary allows flexible protein coding
We had shown that RAC sites were enriched in the -100 nt region of exon start in C1 exons. An open hypothesis is whether a distinct amino acid or codon usage exists in these exons. To test this hypothesis, we counted the codon usage for the first 30 codons (30 ×3 nt = 90 nt) in each exon, and also calculated its corresponding amino acid usage. We found that amino acids D, N, and T were the 3 mostly enriched in last exon of C1, while amino acids of S, P, and A were the 3 mostly avoided (Fig. 7a). Consistent with amino acids usage in last exon, D, N, and T were also enriched in internal exons of C1, while S, P, and A were avoided (Fig. 7b). The strong correlation of odds ratio (C1 vs C2) of amino acids usage (Fig. 7c) supported that last exons and internal exons follow the same amino acid usage bias to effect their m⁶A deposition[23]. As expected, the codons for D, N, T were enriched in C1 internal exons, while codons coding A, S, P were avoided (Fig. 7d, e). Moreover, the

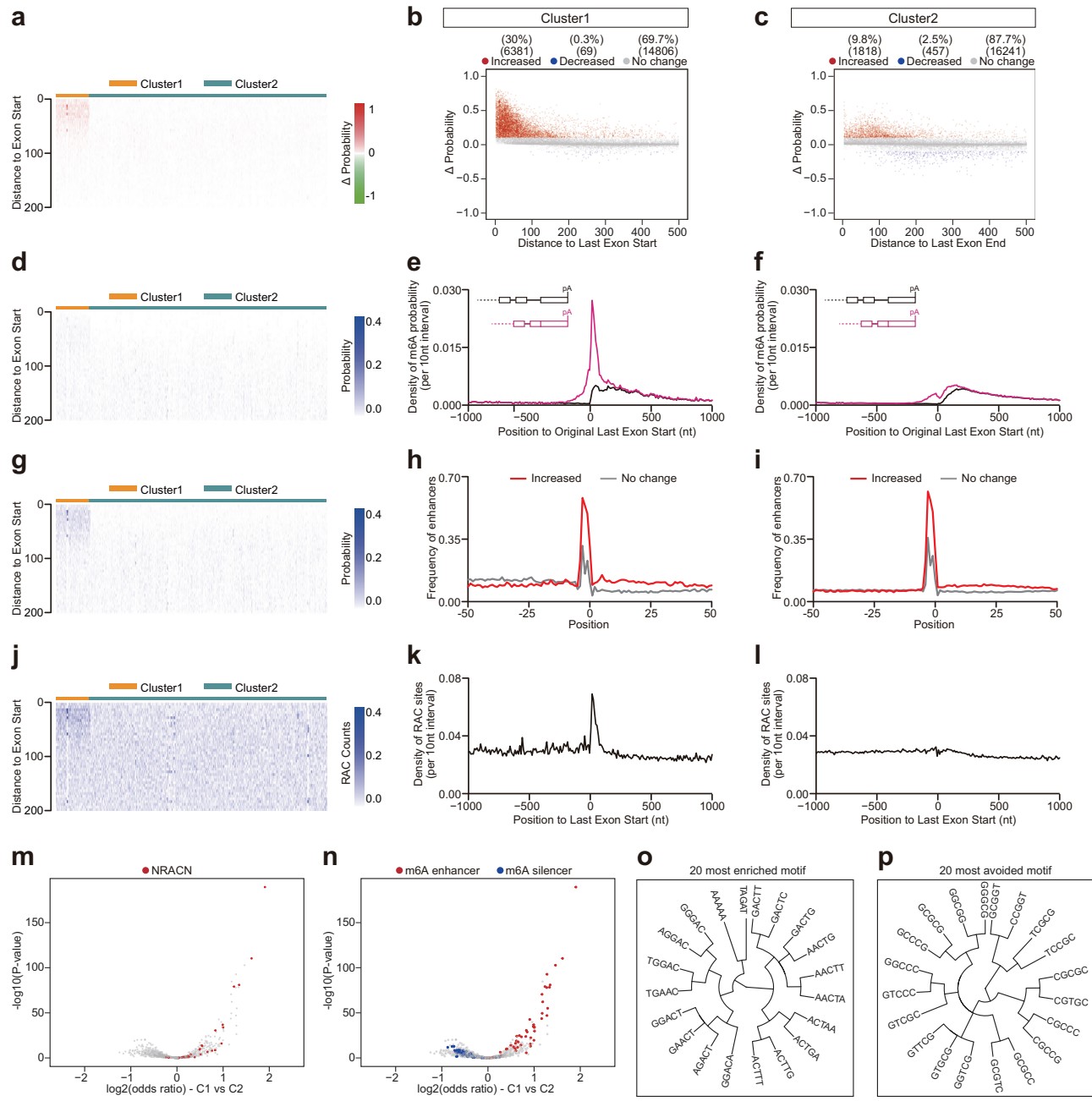

**Fig. 4 | A proportion of last exons exhibit strong m⁶A deposition inhibition by exon-intron boundary. a**, **d**, **g**, **j** The heatmap visualized ΔProbability (**a**), m⁶A Probability (**d**), m⁶A Probability by last intron deletion (**g**), and counts of RAC sites (**j**) in the first 200 nt of last exon. The 200 nt was binned into 40 intervals (5 nt interval). Genes were clustered (see details in Methods) into two clusters (Cluster1, Cluster2) based on ΔProbability. **b**, **c** Positional plot of ΔProbability (**b** for Cluster1, **c** for Cluster2) for the RAC sites located in the first 500 nt region of last exons (*n* = 1500). Red, blue, and gray dots were those sites that had increased (> 0.1), decreased (< −0.1), or not change probability (|ΔProbability| <= 0.1) respectively by last intron deletion. **e**, **f** The m⁶A density around original last exon start (**e** for Cluster1, **f** for Cluster2) were compared for transcripts with full length (black line),

last intron deletion (pink line). **h**, **i** Positional plot for the frequency of top 50 m⁶A enhancers (**h** for Cluster1, Fig. 4i for Cluster2) in mRNA sequences around the RAC sites. Increase sites (red line, ΔProbability > 0.1), and no change sites (gray line, |ΔProbability| <= 0.1). **k**, **l** The density of RAC sites around last exon start (**k** for Cluster1, **l** for Cluster2). **m**, **n** Pentamer enrichment in different last exon start regions comparing Cluster1 vs. Cluster2. The *y*-axis showed the −log10(two-sided Fisher-exact test *P*-value), and the *x*-axis indicated the log2(odds ratio values). In **m**, NRACN motifs were highlighted in red. In **n**, top 50 m⁶A enhancers or silencers were highlighted in red or blue respectively. **o**, **p** Dendrogram showed clustering of 20 most enriched (**o**) or avoided (**p**) motifs comparing Cluster1 vs. Cluster2.

odds ratio (C1 vs C2) of codon usage also had strong correlation between last exon and internal exon (Fig. 7f). We noticed that sets of synonymous codons encoding the same amino acids had quite different codon usages in C1 versus C2 exons. For example, the GAC codon was more frequently used than synonymous codon GAT in C1 exons (Fig. 7g), and AAC codon was also more enriched than synonymous AAT codon (Fig. 7h).

These data suggest that the m⁶A deposition inhibition by exon-intron boundary might allow flexible protein coding that could be needed in the C1 exons. Though these exons contained the biased amino acid and codon usage for specific protein coding and beyond, they didn't appear to have the enriched m⁶A signal due to the m⁶A deposition inhibition by exon-intron boundary. A very interesting question would be which one could come first in evolution: did the

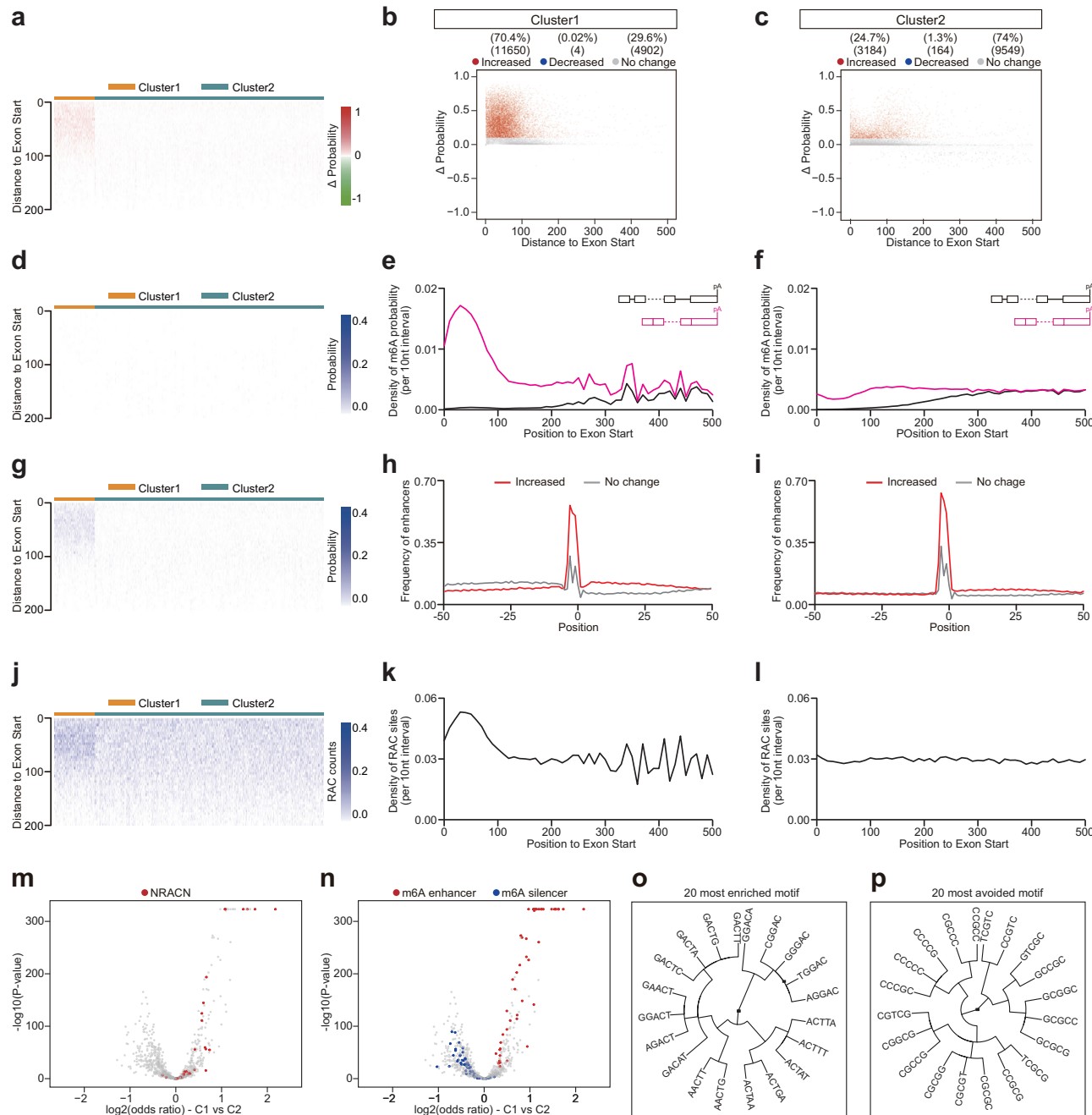

**Fig. 5 | A proportion of internal exons exhibit strong m⁶A deposition inhibition by exon-intron boundary. a**, **d**, **g**, **j** The heatmap visualized ΔProbability (**a**), m⁶A Probability (**d**), m⁶A Probability by introns deletion (**g**), and counts of RAC sites (**j**) in the first 200 nt of internal exon. The 200 nt was binned into 40 intervals (5 nt per interval). Exons were clustered (see details in Methods) into two clusters (Cluster1, Cluster2) based on ΔProbability. **b**, **c** Positional plot of ΔProbability (**b** for Cluster1, **c** for Cluster2) for the RAC sites located in the first 500 nt region of internal exons (*n* = 3000). Red, blue, and gray dots were those sites that had increased (> 0.1), decreased (< −0.1), or not change probability (|ΔProbability| <= 0.1) respectively by introns deletion. **e**, **f** The m⁶A density at internal exon start (**e** for Cluster1, **f** for Cluster2) were compared for transcripts with full length (black line), introns

deletion (pink line). **h**, **i** Positional plot for the frequency of top 50 m⁶A enhancers (**h** for Cluster1, **i** for Cluster2) in mRNA sequences around the RAC sites. Increase sites (red line, ΔProbability > 0.1), and no change sites (gray line, |ΔProbability| <= 0.1). **k**, **l** The density of RAC sites at internal exon start (**k** for Cluster1, **l** for Cluster2). **m**, **n** Pentamer enrichment in different last exon start regions comparing Cluster1 vs. Cluster2. The *y*-axis showed the −log10(two-sided Fisher-exact test *P*-value), and the *x*-axis indicated the log2(odds ratio values). In **m** NRACN motifs were highlighted in red. In **m**, NRACN motifs were highlighted in red. In **n**, top 50 m⁶A enhancers or silencers were highlighted in red or blue respectively. **o**, **p** Dendrogram showed clustering of 20 most enriched (**o**) or avoided (**p**) motifs comparing Cluster1 vs. Cluster2.

splice site evolve first, therefore blocking methylation thus enabling more RAC motifs/codons to appear? or did these methylation sites evolve first, requiring splice sites to come up to inhibit m⁶A deposition and therefore mRNA degradation? Both scenarios could be true and are interesting questions to pursue in natural evolutionary study.

Besides the protein coding bias, we found that the length of C1 internal exons was shorter than C2 internal exons, while the length of its nearby introns including upstream and downstream intron was longer (Fig. 7i). In addition, C1 exons were more likely to be constitutive exons than alternative exons (Fig. 7j).

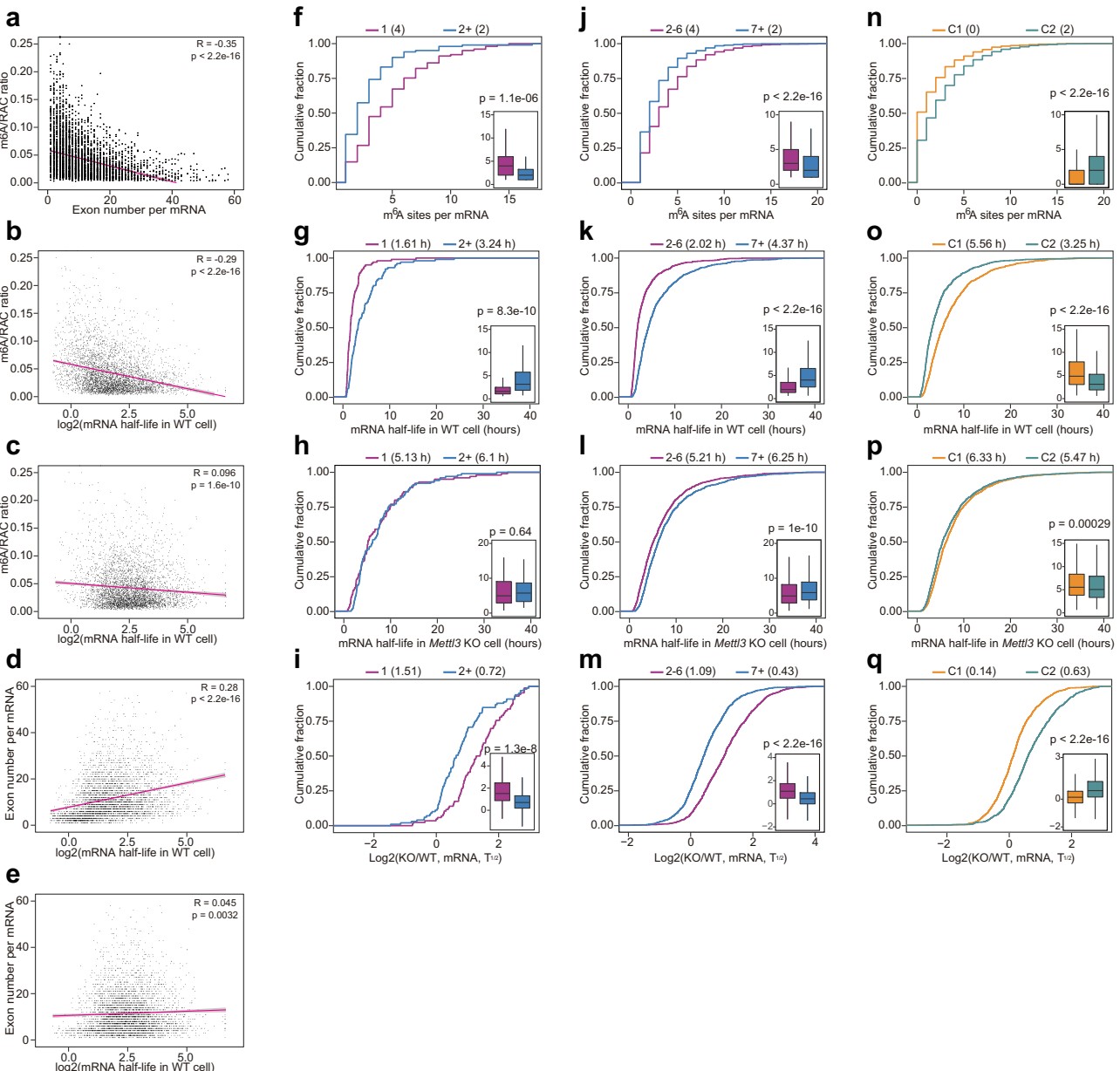

**Fig. 6 | The m⁶A deposition inhibition by exon-intron boundary enables longer mRNA half-lives. a** The scatter plot of m⁶A/RAC ratio for mRNA exon numbers for individual mRNAs (dots), The *P*-value was determined by two-sided Student's *t*-test. **b**, **c** The scatter plot of m⁶A/RAC ratio for mRNA $T_{1/2}$s (**b** for *Mettl3* WT mouse ES cells, **c** for *Mettl3* knockout mouse ES cells) for individual mRNAs (dots). The *P*-value was determined by two-sided Student's *t*-test. **d**, **e** The scatter plot of exon numbers for mRNA $T_{1/2}$s (**b** for *Mettl3* WT cells, **c** for *Mettl3* knockout cells) for individual mRNAs (dots). m⁶A may majorly account for the correlation that multi-exon genes have more stable mRNAs. The *P*-value was determined by two-sided Student's *t*-test. **f–i** Cumulative distribution and boxplots (inset) showing m⁶A sites number (**f**), mRNA $T_{1/2}$s in *Mettl3* WT cells (**g**), mRNA $T_{1/2}$s in *Mettl3* knockout cells (**h**), and mRNA $T_{1/2}$s changes upon global m⁶A loss (**i**) in single (*n* = 101) and multiple exon genes (*n* = 101). The box represents the 1st to 3rd quartile with the median marked

by a horizontal line. The *P*-values were calculated by two-sided Wilcoxon test. **j–m** Cumulative distribution and boxplots (inset) showing m⁶A sites number (**j**), mRNA $T_{1/2}$s in *Mettl3* WT cells (**k**), mRNA $T_{1/2}$s in *Mettl3* knockout cells (**l**), and mRNA $T_{1/2}$s changes upon global m⁶A loss (**m**) in genes with 2–6 exons (*n* = 1104) and genes with >6 exons (*n* = 1104). The box represents the 1st to 3rd quartile with the median marked by a horizontal line. The *P*-values were calculated by two-sided Wilcoxon test. **n–q** Cumulative distribution and boxplots (inset) showing m⁶A sites number (**n**), mRNA $T_{1/2}$s in *Mettl3* WT cells (**o**), mRNA $T_{1/2}$s in *Mettl3* knockout cells (**p**), and mRNA $T_{1/2}$s changes upon global m⁶A loss (**q**) in genes of Cluster1 (*n* = 1104) and genes of Cluster2 (*n* = 1104). The box represents the 1st to 3rd quartile with the median marked by a horizontal line. The *P*-values were calculated by two-sided Wilcoxon test.

In summary, by in silico high-throughput mutational modeling and experimental validations, we found that exon-intron boundary inhibited the m⁶A deposition at nearby exons. The site-specificity of m⁶A deposition were influenced by both local cis-regulatory elements and this exon-intron boundary inhibition mechanism. Our work provides new insights into the mechanism of m⁶A site-specific deposition and its global distributional bias or hallmark (Fig. 7k).

## Exon junction complex partially contributes to m⁶A deposition inhibition by exon-intron boundary

During our manuscript review period, there were three independent papers published online which found that exon junction complex (EJC) could contribute to the exon-intron boundary inhibition of m⁶A[31–33]. In contrast to these three papers which claim that this EJC inhibition is universal for m⁶A inhibition, we found that their EJC depletion/

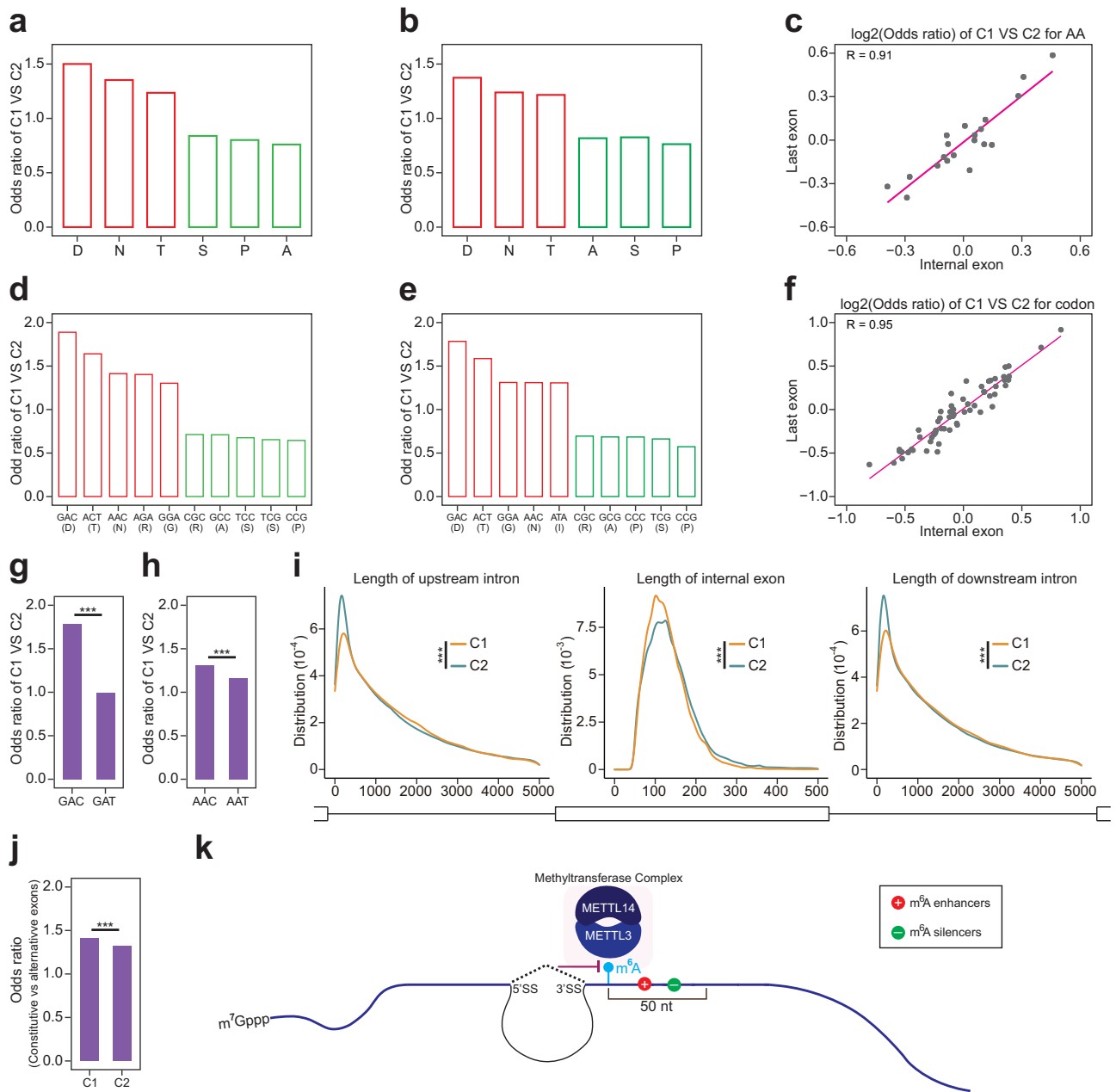

**Fig. 7 | The m⁶A deposition inhibition by exon-intron boundary enables flexible protein coding. a**, **b** Bar plot of odd ratio for amino acids of Cluster1 vs Cluster2 in last exons (**a**) and internal exons (**b**). **c** Scatter plot of the correlation for $\log_2$(odd ratio) of Cluster1 vs Clusters for amino acids in last exons and internal exons. Each gray dot was an amino acid. **d**, **e** Bar plot of odd ratio for codons of Cluster1 vs Cluster2 in last exons (**d**) and internal exons (**e**). Its corresponding amino acids were also labeled. **f** Scatter plot of the correlation for $\log_2$(odd ratio) of Cluster1 vs Clusters for codons in last exons and internal exons. Each gray dot was a codon. **g** Bar plot of odd ratio for codons (GAC, GAT) encoding D amino acid of Cluster1 vs Cluster2 in internal exons. The *P*-values were calculated by two-sided Fisher-exact test ($p < 2.2e^{-16}$. Significance: ***$p < 0.001$). **h** Bar plot of odd ratio for codons (AAC, AAT) encoding N amino acid of Cluster1 vs Cluster2 in internal exons. The *P*-values

were calculated by two-sided Fisher-exact test ($p = 1.3e^{-9}$. Significance: ***$p < 0.001$). **i** Density plot of internal exon length (middle panel), upstream intron length (left panel), and downstream intron length (right panel). The density was compared between Cluster1 and Cluster2. The *P*-values were calculated by the two-sided Kolmogorov–Smirnov test ($p < 2.2e^{-16}$, $p = 6.4e^{-6}$, $p = 4.2e^{-5}$ for left, middle, and right panel respectively. Significance: ***$p < 0.001$). **j** Bar plot of odd ratio for constitutive vs alternative exons in Cluster1 and Cluster2. The *P*-values were calculated by two-sided Fisher-exact test ($p = 0.00000668$. Significance: ***$p < 0.001$). **k** The site-specificity of m⁶A modification is determined by both local *cis*-elements within 50 nt downstream sequence and the intron inhibition to m⁶A deposition at nearby exons.

knockdown data could partially support that m⁶A is inhibited by exon-intron boundary in a proportion of short internal exons. iM6A modeling demonstrated the m⁶A deposition inhibition by exon-intron boundary occurs in both short (<=200 nt) and long (>200 nt) internal exons (Fig. 8a, c), and m⁶A density increases sharply at C1 exons by intron deletion (Fig. 8a, c). On one hand, EJC depletion indeed increased m⁶A modification in some short internal exons particularly

with a stronger increase in C1 short internal exons (Fig. 8b for Y14 depletion, and Supplementary Fig. 19a for siEIF4A3); on the other hand, EJC depletion had little m⁶A signal increase in long internal exons (Fig. 8d for Y14 depletion, and Supplementary Fig. 19b for siEIF4A3), suggesting additional trans-factors yet to be identified. Besides repressing the m⁶A deposition in internal exons, exon-intron boundary also inhibits the m⁶A deposition in the last exons (Fig. 8e). However,

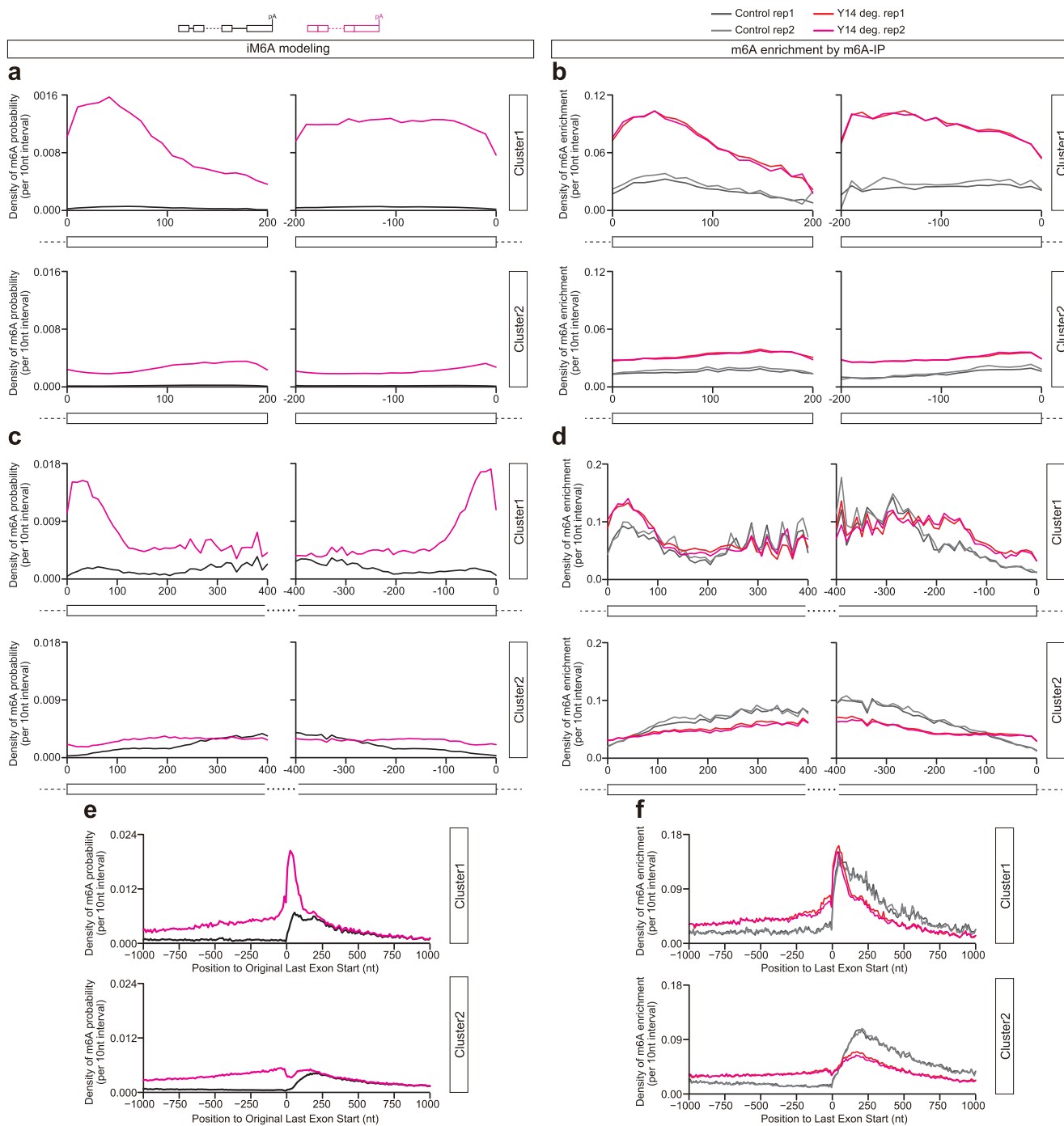

**Fig. 8 | Exon junction complex partially contributes to m⁶A deposition inhibition by exon-intron boundary. a** The m⁶A probability density at short internal exons (<= 200 nt) were compared between full length (black line) and last intron deletion (pink line). These internal exons were divided into two clusters (Cluster1 vs. Cluster2) based on clustering result of Supplementary Fig. 11. **b** The m⁶A enrichment density at short internal exons (<= 200 nt) were compared between control (gray lines) and Y14 depletion (red lines). These internal exons were divided into two clusters (Cluster1 vs. Cluster2) based on clustering result of Supplementary Fig. 11. **c** The m⁶A probability density at long internal exons (> 200 nt) were compared between full length (black line) and last intron deletion (pink line). These internal exons were divided into two clusters (Cluster1 vs. Cluster2) based on clustering result of Supplementary Fig. 11. **d** The m⁶A enrichment density at long internal exons (> 200 nt) were compared between control (gray lines) and Y14 depletion (red lines). These internal exons were divided into two clusters (Cluster1 vs. Cluster2) based on clustering result of Supplementary Fig. 11. **e** The m⁶A probability density around original last exon start were compared between full length (black line) and all introns deletion (pink line). These last exons were divided into two clusters (Cluster1 vs. Cluster2) based on clustering result of Supplementary Fig. 8. **f** The m⁶A enrichment density around last exon start were compared between control (gray lines) and Y14 depletion (red lines). These last exons were divided into two clusters (Cluster1 vs. Cluster2) based on clustering result of Supplementary Fig. 8.

EJC depletion did not affect m⁶A deposition at last exons (Fig. 8f for Y14 depletion, and Supplementary Fig. 19c for siEIF4A3). The loss of EJC could only increase the m⁶A signal on a small proportion of short internal exons (Fig. 8b). Altogether, EJC, as a trans-factor, only contributes to m⁶A inhibition by exon-intron boundary in a small proportion of short internal exons, suggesting that additional factors which may also participate in m⁶A deposition site-specificity are yet to be identified.

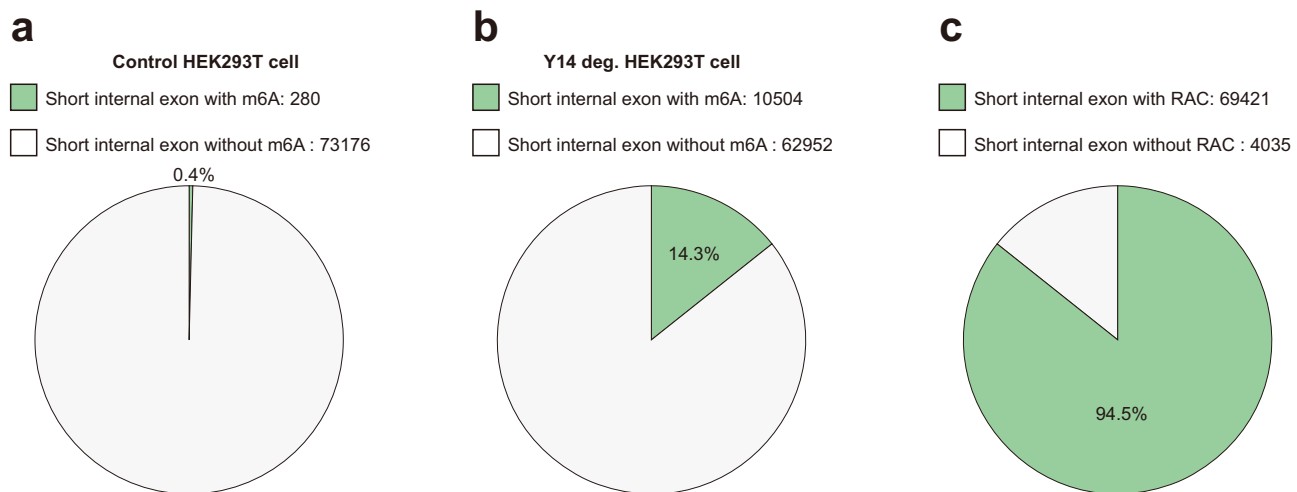

**Fig. 9 | EJC loss increase m⁶A deposition in a subset of short internal exons. a** Pie chart of short internal exons (<= 200 nt) with (280) or without (73176) m⁶A in control HEK293T cell. **b** Pie chart of short internal exons (<= 200 nt) with (10504) or without (62952) m⁶A in Y14 depletion HEK293T cell. **c** Pie chart of short internal exons (<= 200 nt) with (69421) or without (4035) RAC sites.

We examined m⁶A modification in short internal exons. About 0.4% (280 out of 73456 expressed short internal exons) exons had m⁶A modification in control HEK293T cell (Fig. 9a), highlighting that there are m⁶A sites in these short exons escaped exon-intron boundary inhibition. Upon the Y14 EJC component depletion[32], methylated short exons increased to 14.3% (10504 out of 73456) (Fig. 9b). in contrast to the fact that most of short exons were not subjected to EJC inhibition (the actual proportion of short internal exons that have RAC sites is as large as 94.5%) (Fig. 9c). These findings supported that EJC only contributed to m⁶A deposition inhibition in a small subset of short internal exons, and there are m⁶A sites being immune to exon-intron boundary inhibition. Exon-Junction complex (EJC) may only play a partial modulatory rule in inhibiting m⁶A site-specificity and other factors including local *cis*-element environment and more trans-factors involved yet to be discovered.

## Discussion

In this study, we explored the larger scale *cis*-regulatory mechanisms for m⁶A site specificity beyond the local *cis*-regulatory elements. iM6A deep learning modeling showed that exon-intron boundary inhibited a proportion of m⁶A deposition at nearby exons. These findings were supported by experimental validations. Further, we revealed that the m⁶A deposition inhibition by exon-intron boundary exhibited a high degree of heterogeneity in different exons at genomic level, with a strong inhibition in a small group of exons. This m⁶A deposition inhibition by exon-intron boundary allows mRNA with more exons to have longer half-life, and m⁶A is a major contributor to why mRNAs with more exons tend to be more stable. In addition, though some exons have biased amino acid and synonymous codon usage for their specific need for protein coding or beyond, these exons don't appear to have higher m⁶A level due to this m⁶A deposition inhibition by exon-intron boundary.

Our findings that exon-intron boundary inhibited m⁶A deposition at the nearby exonic region close to splice sites and that the repressed m⁶A sites were enriched within the ~100 nt exonic region from either splice site of an exon could help us understand the regional bias for m⁶A modification in mRNAs. Given that most internal exons in vertebrate are short (average size <150 nt)[28], their exonic regions are mostly within the ~100 nt distance to a splice site and hence the m⁶A deposition is inhibited by exon-intron boundary in short internal exons. It could explain why m⁶As are relatively enriched in last exons, as well as long internal exons[20]. As last exon is composed of some coding region and most of the 3'UTR contains >70% of all m⁶A modification in mRNAs[20],

the exon-intron boundary inhibition on m⁶A deposition could focus the concentration of m⁶A signal on last exons and enable the complex and novel 3'UTR regulations involving m⁶A related RNA biology.

It is interesting and important to understand the molecular mechanism how exon-intron boundary inhibits m⁶A deposition. When our manuscript was under review, three independent papers published online reported that exon junction complex (EJC) could contribute to the exon-intron boundary inhibition of m⁶A[31–33], we found that EJC only contributes to the m⁶A inhibition on a small proportion of short internal exons, suggesting additional trans-factors yet to be identified.

Another important question regarding the mechanism of m⁶A deposition is when m⁶A is added to exons. Our previous study demonstrated that m⁶A can be added to exons before the actual splicing cleavage event (e.g. Figure 3 of Ke et al. GD 2017 showed m⁶A deposition to intron-containing exonic region)[11], but the increase of m⁶A deposition by EJC loss suggest that m⁶A can be added to exons after the actual splicing cleavage event. RNA splicing involves multiple steps which include exon/intron definition (i.e. the alpha spliceosome complex), spliceosomes assembly (i.e. the beta spliceosome complex and beyond, steps before the actual splicing cleavage event), two-step splicing reaction (the actual splicing cleavage event), EJC assembly (post the splicing cleavage event)[34]. It is possible that the time range when m⁶A is added to pre-mRNA/mRNA covers the entire time range of pre-mRNA splicing which includes both pre- and post- splicing cleavage event, and the pre-mRNA splicing inhibition on m⁶A may exist in some or all these wide time ranges. Pre-mRNA splicing is a very plausible mechanism by which the exon-intron boundary may influence m⁶A deposition, but other possibilities could be involved. These full mechanism details are all exciting future directions for the field to settle in the years ahead.

Our deep learning modeling approach highlights that the m⁶A deposition site-specificity is overwhelmingly determined by primary nucleotide sequences which includes both local *cis*-element motifs but also long-range *cis*-element regulation such as exon-intron boundary. All these facts support the view that m⁶A is "hard-wired" in the genome by genomic sequences which echoes the view of some other colleagues in the field[8,35] (e.g. the Murakami & Jaffrey review[8] in proposing the gene structure relationship with m⁶A pattern and a potential role, and the He & He review[35] discussed a related view). Given that, the dynamic regulation of m⁶A might not be a phenomenon that could be observed in most m⁶As. It is analogous to the situation of pre-mRNA

splicing that most of pre-mRNA splicing is constitutive splicing though there does exist alternative splicing as a minor group. There might be m⁶A dynamics, as it is hard to rule out this possibility completely; if so, it would be likely to exist in a relatively fewer number compared to the static m⁶A methylation, though the underlying functional importance is yet to be established. In the same vein, alternative splicing regulation is an important layer of tissue-specific gene expression, though its number is much fewer than that of constitutive splicing. As a young field of m⁶A RNA biology, these directions are all exciting future questions of great importance.

Vertebrate genes primarily consist of short exons separated by large introns while lower eukaryotes genes (yeast as an example) are made up of a large number of intronless genes or genes with long exons separated by small introns[36]. In yeast, m⁶A methylation occurs only during meiosis as the METTL3 yeast homolog IME4 expression is only expressed in this time period[37–39]. In mammals, the m⁶A deposition inhibition by exon-intron boundary may allow transcripts to have low methylation level in general despite the widespread expression of METTL3 across different tissues and cell types. In this study, we showed that C1 internal exons exhibit strong m⁶A deposition inhibition by exon-intron boundary. Comparing to other exons, these C1 exons tend to be shorter in length while being flanked longer 5′ and 3′ introns (Fig. 7i), suggesting the exon definition model could play an important role for these C1 exons. Furthermore, the finding that C1 internal exons tend to be constitutive exons not alternative exons (Fig. 7j), suggesting that the robust pre-mRNA splicing efficiency of constitutive exon may contribute to the exon-intron boundary inhibition of m⁶A methylation.

A major function of m⁶A is to promote mRNA decay[9–12]. We demonstrated that the m⁶A deposition efficiency has a strong anti-correlation with pre-mRNA splicing events, and mRNAs with higher exon number have lower m⁶A deposition efficiency. Thus, m⁶A deposition inhibition by exon-intron boundary enables transcripts with multiple exons to have long mRNA half-life. Our work reveals that m⁶A is a major contributor to why mRNAs with more exons tend to be more stable. As this study has shown, in comparison to transcripts with multiple exons, transcripts with single exon have higher m⁶A levels and possess shorter $T_{1/2}$s. Similarly, transcripts with lower exon number have higher number of m⁶A sites, as well as shorter $T_{1/2}$s. Many important regulatory genes are intronless, including many immediate early genes (e.g. c-*Fos* gene) and important transcriptional factors (e.g. *Sox2* gene). The mRNAs of these genes are generally short-lived and have many m⁶As. Being intronless with more methylated sites, this leads to shorter half-life and lower activity, often appropriate for their evolved function to be able to response acutely to rapid environmental perturbations.

It has been well established that pre-mRNA splicing could influence mRNA half-life through the non-sense mediated decay (NMD) pathway[40], and our finding that exon-intron boundary/pre-mRNA splicing inhibited m⁶A deposition to increase mRNA half-life provided a completely new avenue for the regulation of pre-mRNA splicing on mRNA stability.

## Methods

### Modeling m⁶A deposition in pre-mRNA by iM6A
We pulled singularity container (tensorflow-19.01-py2) from NVIDA official website to create the environment for iM6A[23], extra packages including biopython (1.76), scikit-learn (0.20.3), keras(2.0.5) were installed into external path by pip. The gene annotation tables (vM7 for mouse, v19 for human) were downloaded from GENCODE (https://www.gencodegenes.org/), and the longest transcript was extracted for each gene. The nucleotide sequence of pre-mRNA served as input, and the probability of each nucleotide being a m⁶A site was calculated by iM6A (Fig. 1a). For intron deletion, the sequences of the corresponding introns were deleted from the gene, and the m⁶A density

around last exon start was compared between full length transcripts and the intron deletion control. For the RAC sites in exonic regions, the delta changes of m⁶A probability value (ΔProbability) after intron deletion were calculated. Then, the sites were categorized into three groups (increased, decreased and no change) based on ΔProbability (cutoff = 0.1). Positional plot and scatter plot were used to characterize ΔProbability distribution in exons.

### Positional plot of pentamers in sequences flanking m⁶A sites
For the RAC sites in last exon and second-to-last exon, we calculated their m⁶A probability change (ΔProbability) for last intron deletion by iM6A. The sites were categorized into three groups (increase, decrease and no change) based on ΔProbability (cutoff = 0.1). We extracted the 55 nt upstream and downstream sequences flanking the RAC sites in mRNA, and the pentamers were enumerated from the 5′ end to the 3′ end of the sequence. The m⁶A enhancers and silencers were quantified by iM6A through saturation mutation data analysis[23]. For positional plot, we counted the numbers of top 50 enhancers and top 50 silencers at each position of sequence. Then, the frequency of the enhancers or silencers were calculated. The plots were compared between the increased sites and no change sites. Similar strategy was applied to the RAC sites in internal exons.

### Conservation analysis of RAC sites
For the RAC sites in last exon and second-to-last exon, we calculated their m⁶A probability change (ΔProbability) for last intron deletion by iM6A. The RAC sites were categorized into three groups (increased, decreased and no change) based on ΔProbability (cutoff = 0.1). Those sites in degeneration position of synonymous codons were selected, and box plot was used to compare the PhyloP score between increased and no change sites. The *P*-values were determined by Wilcoxon test. Similar strategy was applied to the RAC sites in internal exons.

### Point mutation for 5′ and 3′ splice sites of last intron in pre-mRNA
For multi-exon genes (>=3 exons), its sequences of last introns were truncated to 200 nucleotides by keeping 100 nucleotides of intron start and intron end. Next, the 5′ splice site (donor: GT dinucleotide), 3′ splice site (acceptor: AG dinucleotide) of mini-introns were mutated to CA, TC respectively. In addition, the cryptic splice sites were predicted by SpliceAI[41] for the sequence of second-to-last exon, mutated truncation intron and last exon. All of cryptic splice sites (Probability > 0.1) were also mutated (donor: mutated to CA; acceptor: mutated to TC). Finally, we only kept the genes (*n* = 2370) which had no new cryptic sites after this 1st round of cryptical splice site point mutation according to SpliceAI, and iM6A was used to model the m⁶A deposition.

### Construction of the minigene
The backbone of minigene was a common retroviral GFP vector, and puromycin was the selection marker for stable cell line. *Gne*, and *Lrp12* were used as the two model genes for experimental validation. For each mRNA, the second-to-last exon was truncated to 100 nt by keeping the 100 nt exonic sequence upstream of the exon end, last intron was truncated to 200 nt by keeping the 100 nt intronic sequences at each end of the last intron, and last exon was truncated to 240 nt by preserving the 240 nt downstream of the exon start. The AcGFP1 was in-frame fused to the second-to-last exon. To avoid non-sense mediated decay (NMD) effect, both genes have stop codon in the last exon. The detailed sequence for the *Gne* and *Lrp12* constructs are in the Supplementary Table 1

### mRNA decay assay
The stable cell lines constantly expressing the minigenes were subjected to four time points (0, 3, 6, and 9 h) of post actinomycin D

treatment (final concentration of 1 μg/mL; Sigma, no. A9415) treatment in three biological replicates. Total RNA of each sample was extracted and quantified by qRT-PCR. The normalized mRNA levels at 0 h were set to 100%. The $T_{1/2}$ was determined as $ln(2)/k$, where $k$ is the decay rate constant. The mRNA levels at different time points were fitted to a first-order exponential decay curve to calculate the $k$.

### m⁶A quantification by SELECT method

The constructs of minigenes were transfected to HEK293T, and total RNA was extracted after 48 h. The elongation and ligation-based qPCR amplification method SELECT[30] was used to quantify the m⁶A modification. For each RAC site in mRNA, the $C_t$ value of m6A sites was first normalized to two non-RAC sites at each construct to calculate the m6A signal level for each site; the fold change of intensity for each m6A site was calculated by comparing their normalized $C_t$ value differences for each m6A site between intron-containing and intron-deletion constructs. Oligos are listed in Supplementary Data 1.

### Clustering exons based on ΔProbability of m⁶A by intron deletion

For the RAC sites located in last exons (Fig. 4 for mouse, and Supplementary Fig. 8 for human), we calculated the delta changes of m⁶A probability value (ΔProbability) by last intron deletion. The first 200 nt of last exon was binned into 40 intervals (5 nt per interval). In each interval, the site with maximum of probability change was selected, while its corresponding ΔProbability was kept as the ΔValue for the interval. Exons then were clustered into two clusters (Cluster1: abbreviated C1, Cluster2: abbreviated C2) by k-means method based on the ΔValue. The heatmap visualized ΔValue (Fig. 4a), average m⁶A Probability (Fig. 4d), average m⁶A Probability after last intron deletion (Fig. 4g), and average count of RAC sites (Fig. 4j) in each interval. The same strategy was applied to cluster the internal exons upon all introns deletion (Fig. 5 for mouse, and Supplementary Fig. 11 for human).

### Correlation analysis between m⁶A and exon numbers

For each transcript, the m⁶A sites (Probability > 0.05) were predicted by iM6A, and total number of RAC sites in exons were also counted. Scatter density plot was used to visualize the correlation between m⁶A/RAC ratio and exon numbers (Fig. 6a). The R-value was calculated by Pearson Correlation Coefficient, and P-value was determined by two-sided Student's t-test. In addition, the transcripts were binned based on exon numbers per mRNA, and boxplot was used to show the m⁶A/RAC ratio or m⁶A density (number of m⁶A sites per 100 nt) in each bin (Supplementary Fig. 18a, b).

### Correlation analysis between m⁶A and mRNA half-life

The mRNA half-lives data were downloaded from Gene Expression Omnibus repository under accession no.GSE86336, Scatter density plot was used to visualize the correlation between m⁶A/RAC ratio and mRNA half-lives ($T_{1/2}$) in Mettl3 WT (Fig. 6b) or knockout mouse ES cells (Fig. 6c). Similarly, the correlation between exon numbers per mRNA and mRNA $T_{1/2}$s in Mettl3 WT (Fig. 6d) or knockout cells (Fig. 6e) was plotted. In addition, the transcripts were binned based on exon numbers per mRNA, and boxplot was used to show the mRNA $T_{1/2}$s in Mettl3 WT (Supplementary Fig. 18c) or knockout cells (Supplementary Fig. 18d) for each bin. The R-value was calculated by Pearson Correlation Coefficient, and P-value was determined by two-sided Student's t-test.

### Analysis of mRNA half-lives

The mRNA half-lives was compared for single-exon vs multiple-exons genes (Figs. 6f–i), 2–6 exons vs >6 exons genes (Fig. 6j–m), C1 vs C2 genes (Fig. 6n–q). We matched the exact RAC sites (Fig. 6) or mRNA length (Supplementary Fig. 18) for transcripts, cumulative distribution and boxplots were used to show m⁶A sites number, mRNA $T_{1/2}$s in

Mettl3 wild-type (WT) cells, mRNA $T_{1/2}$s in Mettl3 knockout (KO) cells, and mRNA $T_{1/2}$s changes upon global m⁶A loss. Median and inter-quartile ranges were presented for the box plot. The P-values were calculated by Wilcoxon test.

### Comparison of amino acids or codons for C1 vs C2 exons

For the amino acids or codons in last exons or internal exons, we counted the number for each amino acid or codon. Only the genes expressed in mESCs were used (GSE86336). The frequency of amino acid or codon in C1 or C2 exons was calculated, and odd ratio of C1 vs C2 was computed. Fisher-exact test was used to evaluate the significance. Scatter plot was used to visualize the correlation of odds ratio between last exon and internal exon. The R-value was calculated by Pearson Correlation Coefficient.

### Analysis of m⁶A-IP data

We downloaded raw sequencing data from Gene Expression Omnibus (GEO) repository (GSE204980, GSE207663). Raw sequencing data was mapped to the hg19 reference genome by bowtie2. For further analysis, the BAM files were filtered for uniquely aligned reads. The read coverage at each nucleotide position to library size was normalized, Then, m⁶A-IP enrichment value was calculated by dividing the normalized read density for m⁶A-IP to that of the input. Positional plot was used to characterize the density of enrichment in exons (Fig. 8). For peak calling (Fig. 9), we searched enriched m⁶A region by scanning the genome with 20 nt sliding windows. The statistical significance of enrichment was calculated by Fisher's exact test (m⁶A-IP vs. input). Benjamini-Hochberg was applied to calculate the FDR for multiple testing. m⁶A-enriched windows were filtered based on enrichment fold (>2) and FDR (<0.05). Then, m⁶A-enriched windows were concatenated for peak with at least 40 nt. The FPKM (fragments per kilo base per million mapped reads) value for each transcript was calculated based on input of m⁶A-IP data, and expressed genes were selected (FPKM >= 1).

### Reporting summary

Further information on research design is available in the Nature Portfolio Reporting Summary linked to this article.

## Data availability

The data supporting the findings of this study are available from the corresponding authors upon reasonable request. The mRNA half-lives data were downloaded from the Gene Expression Omnibus repository under accession no.GSE86336. m⁶A-IP data were downloaded from the Gene Expression Omnibus repository under accession no. GSE204980, and no.GSE207663. Source data for the figures and supplementary figures are provided as a Source Data file. Source data are provided with this paper.

## Code availability

The source code of the manuscript is available at GitHub (https://github.com/ke-laboratory/iM6A-Splicing).

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

## Acknowledgements

We thank Dennis Weiss and members of Ke Laboratory and Ying Laboratory for comments, suggestions, and thoughtful discussions. Ke Laboratory and this research is funded by NIH/NIGMS Maximizing Investigators' Research Award (MIRA) R35 Award (R35 GM133711 to S.K.), American Cancer Society Pilot Award (ACS-2019-Pilot-Ke/IRG-16-191-33/IRG-21-136-36-IRG to S.K.) and the Jackson Laboratory Cancer Center New Investigator award from the NIH/NCI Cancer Center Support Grant (2 P30 CA034196-34 to S.K.).

## Author contributions

S.K., Z.L., and Z.Y. conceived and designed the study. Z.L. conducted the experiments and performed the data analysis. Q.M., S.S., N.L., and H.W. contributed to the test of experimental validation. S.K. and Z.L., wrote the manuscript. S.K. supervised the research.

## Competing interests
The authors declare no competing interests.
