## [Peer Review File · Nature Communications]

Exon-intron boundary inhibits m6A deposition, enabling m6A distribution hallmark, longer mRNA half-life and flexible protein codingREVIEWER COMMENTS

Reviewer #1 (Remarks to the Author):

This manuscript addresses the question of why m6A is enriched in two particular regions: in long internal exons, and in terminal exons. The authors use a deep learning prediction model which they believe demonstrates that splicing inhibits m6A, particularly in the 100 nucleotide region around splice sites, and experimentally validate this, to some degree, using a reporter. Furthermore, the authors also show that m6A is a major contributor to why genes with more exons are more stable.

This manuscript helps to understand the genome features that influence m6A deposition. Furthermore, this manuscript strongly demonstrates the robustness of the M6Ai prediction model, and explores the ability to use this prediction model as a discovery tool for the rules of m6A deposition.

A major problem is that the manuscript is written as though pre-mRNA splicing regulates m6A. Although this is compatible with the results that are presented, the deep learning model cannot distinguish between whether m6A appears before or after splicing. In the manuscript, the authors discuss m6A as though m6A tells them that the methylation occurs after splicing. However, I don't see evidence that the regulation is not via the exon boundaries, rather than splicing itself. Consistent with this is the authors' previous work which showed that m6A was co-transcriptional and methylation occurred on mRNAs containing introns, and was therefore occurring prior to splicing. Thus, unless I have misunderstood the data, substantial re-writing is needed to take away the "temporal" aspect of their model, i.e., methylation that occurs or is regulated by a splicing event (and thus occurs after splicing), and simply refer to methylation with respect to exon boundaries.

This manuscript relies heavily on the prediction model, and only uses a few experiments to confirm their models. However, a few papers were very recently published which are consistent with the authors' models: Yang et al, 2022, Nat Comms (DOI: 10.1038/s41467-022-35643-1), Uzonyi et al, 2023, Mol Cell (DOI:10.1016/j.molcel.2022.12.026), He et al, 2023, Science (DOI: 10.1126/science.abj9090). These recent manuscripts complement the manuscript under review. However, the authors should do more to highlight the differences in their interpretations compared to the published work.

My specific concerns are enumerated below:

Major concerns

1. The concept of "m6A enhancers" or "silencers" is not very clear. My understanding is that these are an unknown elements which are defined by the machine learning model, but a clearer explanation may help the reader follow the text better. One thing which was not clear to me was whether DRACH motifs are included under m6A enhancers. Additionally, perhaps the authors can speculate what these enhancers or silencers are. One possibility is that splice sites are silencers – is it possible for the authors to test this?

2. The authors often state that removing splicing increases m6A in a 100nt region around the splice site. However, the majority of exons are <200nt, hence increase in m6A anywhere in these exons would appear as m6A increasing only in the 100 – 200nt region. To remove this confounding factor, authors should split up short and long exons for all such analysis in the paper (Fig 1b, 1c, 1i, 1h, 2a, 2b, 2c, 2h, 2i, 4b, 4c, 4e, 4f, 5b, 5c, 5e, 5f, and corresponding supplementary figures). This could clarify whether the increase in m6A is an overall increase in m6A in short exons, or that m6A is specifically increased near splice sites, even in long exons. For example, due to this confusing effect, Fig 1h does not appear to agree with Fig 1i. Fig 1h shows a sharp increase in m6A near the second last exon start, but Fig 1i seems to show this occurs near the exon end. Splitting the short and long exons up may more clearly demonstrate that m6A increases near the second last exon end for all exons, and the effect in 1i is primarily due to the majority presence of short exons.

3. As stated above, the authors claim that “pre-mRNA splicing inhibits m6A deposition”, implying causation. It is not well-established through this study that there is a causative relationship between pre-mRNA splicing and m6A deposition. Due to the “black box” of machine learning, it is not entirely clear what the relationship between splicing and m6A deposition is. One major issue with the causation model is that m6A is thought to occur co-transcriptionally, i.e. before or during splicing. The author had previously established that m6A is found on unspliced transcripts (Ke et al, 2017, *Genes and Dev*), implying that m6A comes before splicing. However, if splicing inhibits m6A deposition, splicing must occur before m6A deposition. The temporal relationship is therefore unclear and contradictory. The authors should comment on their previous work, and the order of events if they think splicing is inhibitory to m6A deposition. It is much better to refer to exon boundaries rather than a causative event, i.e., pre-mRNA splicing.

4. There are a few recent papers which discuss a similar phenomenon of splicing inhibiting m6A: Yang et al, 2022, *Nat Comms* (DOI: 10.1038/s41467-022-35643-1), Uzonyi et al, 2023, *Mol Cell* (DOI:10.1016/j.molcel.2022.12.026), He et al, 2023, *Science* (DOI: 10.1126/science.abj9090). I suggest that the authors include these in their introduction, throughout the results section (for comparisons) and in their discussion. These manuscripts are complementary to the authors’ data and support the authors’ model that splicing inhibits m6A deposition.

5. Can the authors comment on whether the majority of “exon heterogeneity” is mostly attributed to DRACH density, or if there are additional factors? For example, the authors can characterize the exon lengths of cluster 1 and cluster 2 (Fig 4), which are known to be a large contributing factor to m6A deposition. Additionally, can the authors comment on the 3’ UTR length, which is thought to be positively associated with m6A?

6. The authors show that latent m6A sites are more conserved than no-change sites. These latent m6A sites are often near splice sites. It is possible that these splice sites are conserved due to splicing, rather than m6A-related biology. Can the authors show a side-by-side comparison of the conservation of all splice sites to see how much the m6A machinery contributes to the conservation compared to the

contribution of the splicing machinery?

7. The authors suggest that C1 exons need to be hidden from m6A as they encode a particular set of codons, many of which look like the “RAC” motif (amino acids D, N, and T). I wonder which came first evolutionarily. Did the splice site evolve, therefore blocking methylation at these sites, allowing more RAC motifs/codons to appear, or did these methylation sites evolve first, requiring splice sites to cover to prevent m6A deposition and therefore mRNA degradation?

8. To me, one of the most fascinating aspects of the study, which also distinguishes it from the other competing published papers, is that this manuscript is very clear that only a subset of sites are influenced by splicing. It would be great to document this even more. Some -simple- experiments could include showing some individual examples of splicing junctions where there is an adjacent DRACH (GGACU) sequence, yet it is highly methylated. A broader computational experiment could involve calculating the number of DRACH sites within 10 nt or so of an exon junction that have “above-median” m6A stoichiometry (or a similar analysis). This can really document the prevalence of DRACH sites that escape. One thing that I would suggest that the authors consider is whether the manuscript could be re-pitched to show that DRACH sites near exon junctions are in fact, NOT, always suppressed. Since there are three competing manuscripts, the authors may consider the unique strengths of this manuscript, which highlights that the EJC suppressor model does not fully explain m6A stoichiometry. To me, showing that exon junction complexes are modulatory, but not fully determinative, is a very important advance in the field.

Related to this, can the authors speculate why some exon junctions are repressors for m6A deposition while others are not? This will be asked by most readers of this paper.

9. The authors argue that internal exon methylation and terminal exon methylation is similar. However, I believe this is not correct. The long internal exons have high methylation throughout the internal region of the long exon. However, the terminal exon only has methylation ~100-200 nt from the beginning of the exon, and then lack methylation throughout the remainder of the terminal exon (which the authors previously showed is typically long). Doesn't this demonstrate very different properties and mechanisms for terminal exons?

10. For many readers the im6A deep learning method is not clear, and it is not clear how the authors can remove introns/exons and model different patterns of methylation. Schematics and figures would be helpful, perhaps to start the manuscript.

11. I don't believe that the one wet experiment supports the model - note that in Fig 3C, A1 goes up after intron removal and A4 does not. This does not make sense. In Fig 3D, shouldn't A3 go up with exon removal? Perhaps the SELECT assay is noisy, but the results were not completely helpful. Comments on the discrepancies and explanations are needed.

Minor concerns and suggestions

1. In Fig 1 (and Fig 2), it is ambiguous what authors mean by “last exon start”, as the “last exon start” is different before and after mutation. They should use “original last exon start” or some other naming convention.

2. The authors should be more clear about what “the gene” (line 100) refers to. Are they performing a transcriptome-wide in silico mutation of every last intron, or on a specific gene?
3. The schematic of the gene in Fig 2a makes the pink gene appear to have all introns except the last two introns.
4. The authors should mention the original mean intron length as a comparison to the truncated intron length of 400 nt (line 170)
5. Line 208 has a typo: “experimentally” instead of “experientially”
6. Fig 4a, d, g and j are missing Y-axis labels, and labels for the colorbar. Figs 4b, c, e, f, h, i, k and l should also be labelled with “cluster 1” or “cluster 2” to improve readability.
7. The correlations in fig 6b, c, d and e are not very apparent due to the scale of changes. Plotting the log-scale half-life may show these correlations more clearly.
8. I think that some people are going to wonder why the initial studies by Ke seeking to identify adjacent sequences that influence m6A deposition did not come up with exon junction regions/sequences/motifs as de-enriched. It might be interesting to talk about why this did not come up in earlier work by Ke, and possibly others. It is possible that it was actually seen by other groups or by Ke, but they did not focus on it. If so, it would be interesting to mention if it is present in other people’s data sets.
9. It may be worth citing the recent Murakami review in Molecular Cell that argued that “gene architecture” would regulate m6A deposition based on their analysis of the commonality of m6A sites in different maps, as well as the relationship between m6A and splice junctions. Do the authors’ data concur with their argument that m6A maps between tissues should be very similar and that m6A is “hard-wired” by gene architecture? Does the authors’ data support the idea that m6A is unlikely to be dynamic? It would be useful to touch on these issues in the Discussion.

Reviewer #2 (Remarks to the Author):

In this manuscript, the authors utilized the deep learning method to model how splicing influences m6A deposition and revealed the pre-mRNA splicing inhibits m6A deposition. This explains why m6A is enriched in last exons and long internal exons. Experiments were well designed to confirm the findings of modeling, and m6A sites in minigenes were examined with single nucleotide resolution. In general, the manuscript is well-written, and shows broad interest for the field of m6A.

1. For last intron deletion, only 4.3% of sites in the last exons showed increased m6A deposition. Overall, the sites in the distal region did not have m6A deposition changes. This makes the inhibitory roles of last intron unreal. To be more specific, the percentage can be calculated for the sites close to last exon start.

2. Due to the fact that m6A is enriched in long exons, do the last exons with different lengths show the same m6A deposition inhibition by the last intron?
3. In Figure 2, do the long internal exons show the same m6A deposition inhibition by intron splicing as short internal exons?
4. In the legends of Figure 2, a typo occurred, and the panels were not correctly labeled.

Reviewer #3 (Remarks to the Author):

In this work Lou et al investigate the relation between RNA splicing and m6A deposition. In their intro they lay out some of the disagreement in the field around this interesting topic of research (p.2 line 47-57). Several works noted the association between splicing and m6A with m6A appearing to lack in internal exons but not in the last exon or long internal exons. While some labs claimed that m6A may be affecting/regulating splicing, the authors propose a different hypotheses by which splicing inhibits m6A. In this work the authors lay out several claims to support this hypothesis.

Most of the work presented is based on computational analysis using the authors previously published iM6A algorithm (Lou et al Nat Comm 2022). iM6A is a CNN based DL model that using a window of genomic sequence to classify a location as an m6a site. In the original publication they authors preformed saturation mutagenesis analysis to find positions downstream of the site were important for the classification. Here they instead replaced the upstream sequence introns with exons and find a large effect. They then go on to extract various statistics based on that, before turning to experimental validation. The validation involved a mini-gene reporter assay and two constructs - one for Lrp12 and another with Gne. The authors compute statistics over the identified sites in these areas and show some of them change between constructs that have the introns and those where the introns were removed.

Overall we find the topic very interesting and we appreciate the work that went into this project. However, we have major concerns with the results and associated claims, as presented.

One fundamental issue we found is that the main conclusions are well supported by the data/analysis. The association between splicing and m6A deposition is definitely intriguing, as is the suggestion that the causal relationship is splicing affects m6A deposition. However, the authors ignore the fact that a common factor/mechanism may affect both rather than one affect the other. Regardless, we are quite suspicious of the computational analysis as performed here. The model used here is the same as their previous publication but previously they only identified a strong effect of downstream sequence. This fact is somewhat glossed over. Why is that? One main reason is likely the analysis procedure - single nucleotide variation (original paper) vs. replacing whole regions (current manuscript). However, this kind of analysis is highly problematic: The authors completely ignore the fact that the model they use is a discriminative model trained on genomic sequence. By introducing large changes of sequences (whole exons/introns) they present to the model out of distribution sequences for which results may be completely wrong. Think of showing an image classifier with cats and dogs and then asking it to classify an image of a raccoon. They did not train on such sequences/raccoons. Thus, this reviewer remains

highly skeptical about any downstream analysis presented here as “proof” while all we see are associations based on out of distribution sequences.

Related to that is the lack of sufficient experimental evidence. First, the experiments are done on only two sequences, which is hardly a proof for a global phenomena. Second, the results even on those sequences are not very convincing. Only a few sites change significantly and there is no clear pattern for those that change/not. Last but not least, the experimental design seems flawed: At the very least the authors should have tested a construct with the same/similar sequences genomic sequences w/wo the splicing occurring (e.g. mutating the splice sites).

In summary, we found the claim interesting, the topic important and we appreciate the amount of work that went into this manuscript. However, as a computational paper there is no novelty (published model, just a different analysis using it) and as a manuscript based on novel results we don't believe the conclusions are well supported by the data/analysis.

REVIEWER COMMENTS

Reviewer #1 (Remarks to the Author):

This manuscript addresses the question of why m6A is enriched in two particular regions: in long internal exons, and in terminal exons. The authors use a deep learning prediction model which they believe demonstrates that splicing inhibits m6A, particularly in the 100 nucleotide region around splice sites, and experimentally validate this, to some degree, using a reporter. Furthermore, the authors also show that m6A is a major contributor to why genes with more exons are more stable.

This manuscript helps to understand the genome features that influence m6A deposition. Furthermore, this manuscript strongly demonstrates the robustness of the M6Ai prediction model, and explores the ability to use this prediction model as a discovery tool for the rules of m6A deposition.

A major problem is that the manuscript is written as though pre-mRNA splicing regulates m6A. Although this is compatible with the results that are presented, the deep learning model cannot distinguish between whether m6A appears before or after splicing. In the manuscript, the authors discuss m6A as though im6A tells them that the methylation occurs after splicing. However, I don't see evidence that the regulation is not via the exon boundaries, rather than splicing itself. Consistent with this is the authors' previous work which showed that m6A was co-transcriptional and methylation occurred on mRNAs containing introns, and was therefore occurring prior to splicing. Thus, I unless I have misunderstood the data, substantial re-writing is needed to take away the "temporal" aspect of their model, i.e., methylation that occurs or is regulated by a splicing event (and thus occurs after splicing), and simply refer to methylation with respect to exon boundaries.

This manuscript relies heavily on the prediction model, and only uses a few experiments to confirm their models. However, a few papers were very recently published which are consistent with the authors' models: Yang et al, 2022, Nat Comms (DOI: 10.1038/s41467-022-35643-1), Uzonyi et al, 2023, Mol Cell (DOI:10.1016/j.molcel.2022.12.026), He et al,

2023, Science (DOI: 10.1126/science.abj9090). These recent manuscripts complement the manuscript under review. However, the authors should do more to highlight the differences in their interpretations compared to the published work. My specific concerns are enumerated below:

We highly appreciate Reviewer #1's recognition of our work. We thank Reviewer #1's valuable comments which improved our manuscript. We have carefully revised the manuscript according to the comments. Below is our point-to-point response. The reviewer's comments are in blue, our responses are in black.

Major concerns

1. The concept of "m⁶A enhancers" or "silencers" is not very clear. My understanding is that these are unknown elements which are defined by the machine learning model, but a clearer explanation may help the reader follow the text better. One thing which was not clear to me was whether DRACH motifs are included under m⁶A enhancers. Additionally, perhaps the authors can speculate what these enhancers or silencers are. One possibility is that splice sites are silencers – is it possible for the authors to test this?

Thanks for the questions. In our previous publication of the iM⁶A deep learning modeling (Luo et al, 2022, Nat Comms, DOI: 10.1038/s41467-022-30209-7), we implemented a high-throughput *in silico* saturated point mutations around m⁶A sites and discovered that the local cis-elements that regulating m⁶A site-specificity are highly enriched in the downstream 50 nt region. Furthermore, from such an over one million point-mutation modeling events, we calculated the quantitative contribution to m⁶A site-specificity by each of the total 1024 pentamers using a linear regression model: m⁶A enhancers are identified as top ranked 5mers (i.e. enhancing m⁶A deposition) while m⁶A silencers are identified as bottom ranked 5mers (i.e. silencing m⁶A deposition). Their designated function was validated by independent experimental data (Fig. 3 in Luo et al, 2022, Nat Comms, DOI: 10.1038/s41467-022-30209-7). To make it more clear for readers, we have revised the manuscript for a better explanation.

In our previous work (Fig. 2 of Luo et al, 2022, Nat Comms, DOI: 10.1038/s41467-022-30209-7), we found that many m⁶A enhancers largely include part of RRACH motif. To demonstrate that DRACH motifs are included in m⁶A enhancers, we counted the number of DRACH or RRACH in m⁶A enhancer motifs (**Response Table 1**) and found that indeed most of DRACH motifs were considered as m⁶A enhancers (it is consistently true regardless we consider top50, top150, top250, and top350 pentamers as m⁶A enhancer motifs).

	Mouse				Human			
	Top50	Top150	Top250	Top350	Top50	Top150	Top250	Top350
DRACH (N = 18)	7	9	12	15	8	9	14	15
RRACH (N=12)	6	8	8	8	7	8	9	10

Response Table 1. Number of motifs containing DRACH, and RRACH in m⁶A enhancer motifs.

To examine the possibility of splice sites as m⁶A silencers, we plotted the distribution of m⁶A enhancers/silencers in exons and introns. The m⁶A silencers were enriched at both splice sites (donor and acceptor), while m⁶A enhancers avoided the splice sites (**Response Figure 1**). We have included this figure in **Supplementary Fig.6**.

Response Figure 1. (a-b) Positional plot for the frequency of top 50 enhancers (red line) or silencers (green line) in the sequences of the mini-intron truncated to 200 nt (by keeping 100

nucleotides of intron start and intron end) and last exon (Response Fig. 1a for mouse, Response Fig. 1b for human).

2. The authors often state that removing splicing increases m6A in a 100nt region around the splice site. However, the majority of exons are <200nt, hence increase in m6A anywhere in these exons would appear as m6A increasing only in the 100 – 200nt region. To remove this confounding factor, authors should split up short and long exons for all such analysis in the paper (Fig 1b, 1c, 1i, 1h, 2a, 2b, 2c, 2h, 2i, 4b, 4c, 4e, 4f, 5b, 5c, 5e, 5f, and corresponding supplementary figures). This could clarify whether the increase in m6A is an overall increase in m6A in short exons, or that m6A is specifically increased near splice sites, even in long exons. For example, due to this confusing effect, Fig 1h does not appear to agree with Fig 1i. Fig 1h shows a sharp increase in m6A near the second last exon start, but Fig 1i seems to show this occurs near the exon end. Splitting the short and long exons up may more clearly demonstrate that m6A increases near the second last exon end for all exons, and the effect in 1i is primarily due to the majority presence of short exons.

We appreciate the suggestion which greatly helps in clarifying the finding. Same as the questions 2 & 3 of Review #2, we split the exons into three groups based on its lengths (\leq 200 nt, 200-400 nt, and \geq 400 nt). For last exons, the increased m⁶A deposition were enriched in the 100 nt region to last exon start for all three groups (**Supplementary Fig. 2**), showing m⁶A deposition inhibition by pre-mRNA splicing occurs near the splicing sites for both short and long exons. For short internal exons (\leq 200 nt), the increase of m⁶A deposition by introns deletion appeared in the 100 nt region to both exon ends (**Fig. 2d,e, and Supplementary Fig. 4d,e**). For long internal exons (200-400 nt, and \geq 400 nt), the increased sites were also enriched in the 100 nt region to both exon ends (**Fig. 2f-i, and Supplementary Fig. 4f-i**). To be comprehensive, we split up exons according to their length for all such analysis (**Fig. 2, and Supplementary Fig. 2, 3, 4, 9, 11, 15**).

In the last intron *in silico* deletion modeling, we examined the m⁶A change situation in the second-to-last exon to demonstrate that the inhibitory effect of pre-mRNA splicing exists locally in the 100nt splice-site-adjacent exonic region of the two flanking exons (i.e. second-to-last exon and last exon). To have a better presentation, we updated a new **Fig. 1g**

(corresponding to the original Fig. 1h) which showed the increase of m⁶A deposition (due to the deletion of last intron) occurred only locally in the second-to-last exon as well as last exon (i.e. the mRNA region on the right side of the figure---after the original start of the second-to-last exon which is set to 0 point), without affecting the upstream exons (i.e. little change in the mRNA region on the left side of the figure---before the original start of the second-to-last exon that is set to 0 point). Furthermore, we did the positional scatter plot for RAC sites that only located in second-to-last exon (**Fig. 1h** which corresponds to the original Fig. 1i), it demonstrated that the increased sites were enriched in the 100 nt region close to exon end, again highlighting the local inhibition of m⁶A deposition by pre-mRNA splicing. We also split up the second-to-last exons according to its exon length, the increased m⁶A modification of sites were consistently enriched in the 100 nt region close to second-to-last exon end regardless of the exon length of the second-to-last exon (**Supplementary Fig. 3**). Taken together, all these data strongly indicate that the inhibitory effects of pre-mRNA splicing on m⁶A deposition are highly local, concentrating in the 100 nt exon region right adjacent to splice sites (i.e. the 100 nt exon ending region in second-to-last exon and the 100 nt exon starting region in last exon).

3. As stated above, the authors claim that “pre-mRNA splicing inhibits m⁶A deposition”, implying causation. It is not well-established through this study that there is a causative relationship between pre-mRNA splicing and m⁶A deposition. Due to the “black box” of machine learning, it is not entirely clear what the relationship between splicing and m⁶A deposition is. One major issue with the causation model is that m⁶A is thought to occur co-transcriptionally, i.e. before or during splicing. The author had previously established that m⁶A is found on unspliced transcripts (Ke et al, 2017, *Genes and Dev*), implying that m⁶A comes before splicing. However, if splicing inhibits m⁶A deposition, splicing must occur before m⁶A deposition. The temporal relationship is therefore unclear and contradictory. The authors should comment on their previous work, and the order of events if they think splicing is inhibitory to m⁶A deposition. It is much better to refer to exon boundaries rather than a causative event, i.e., pre-mRNA splicing.

We agree with the reviewer that this is an important question regarding the mechanism of

m⁶A deposition. Previously we found that m⁶A is deposited to pre-mRNA during transcription (Ke et al, 2017, Genes Dev, DOI: 10.1101/gad.301036.117), raising the question of the potential crosstalk between m⁶A deposition and pre-mRNA splicing which is also a co-transcriptional RNA processing event. Our previous study demonstrated that m⁶A can be added to exons before the actual splicing cleavage event (i.e. we showed that the m⁶A containing RNA fragment can contain unspliced intron in the Figure 3 of Ke et al, 2017, Genes Dev, DOI: 10.1101/gad.301036.117). However, RNA splicing involves multiple steps (a good review could be Black DL 2003, <https://doi.org/10.1146/annurev.biochem.72.121801.161720>) which include exon/intron definition (i.e. the alpha spliceosome complex), spliceosomes assembly (i.e. the beta spliceosome complex and beyond, steps before the actual splicing cleavage event), two-step splicing reaction (the actual splicing cleavage event), EJC assembly (post the splicing cleavage event). It's possible that the time range when m⁶A is added to pre-mRNA/mRNA covers the entire time range of pre-mRNA splicing which includes both pre- and post- splicing cleavage event. These full mechanism details are all exciting future directions for the field to settle in the years ahead. Our current work in this manuscript which supports the causality that pre-mRNA splicing inhibits m⁶A deposition include both the *in silico* deep learning modeling that the loss of pre-mRNA splicing enhance the m⁶A deposition at the 100 nt splice-site adjacent exonic region and the followed experimental validation that mRNAs have more m⁶A deposition signal when generated without the involvement of pre-mRNA splicing than mRNAs with identical primary nucleotide sequence but generated undergoing the pre-mRNA splicing. The only known mechanism that could define exon-intron boundaries is pre-mRNA splicing, and many steps of splicing could possibly affect m⁶A deposition which could happen either before or after the actual splicing cleavage. We appreciate the reviewer's suggestion. To better communicate with readers and be open-minded, we update our manuscript title to **“Pre-mRNA splicing/exon-intron boundary inhibits m⁶A deposition, enabling m⁶A distribution hallmark, longer mRNA half-life and flexible protein coding”**.

4. There are a few recent papers which discuss a similar phenomenon of splicing inhibiting

m6A: Yang et al, 2022, Nat Comms (DOI: 10.1038/s41467-022-35643-1), Uzonyi et al, 2023, Mol Cell (DOI:10.1016/j.molcel.2022.12.026), He et al, 2023, Science (DOI: 10.1126/science.abj9090). I suggest that the authors include these in their introduction, throughout the results section (for comparisons) and in their discussion. These manuscripts are complementary to the authors' data and support the authors' model that splicing inhibits m6A deposition.

We thank the reviewer for the suggestions. We re-analyzed the m⁶A-IP in control or EJC depletion cell (Yang et al, 2022, Nat Comms, DOI: 10.1038/s41467-022-35643-1, Uzonyi et al, 2023, Mol Cell, DOI:10.1016/j.molcel.2022.12.026), and found that their EJC depletion data could partially support our finding of m⁶A inhibition by pre-mRNA splicing. iM6A modeling demonstrated the m⁶A deposition inhibition by pre-mRNA splicing occurs in both short (\leq 200 nt) and long ($>$ 200 nt) internal exons (**Fig. 8a,c**), and m⁶A density increases sharply at C1 exons by intron deletion (**Fig. 8a,c**). On one side, EJC depletion indeed increased m⁶A modification in some short internal exons and particularly showed a stronger increase in C1 short internal exons (**Fig. 8b, and Supplementary Fig.19a**), on the other side, EJC depletion had little m⁶A signal increase in long internal exons (**Fig. 8d, and Supplementary Fig. 19b**), suggesting additional trans-factors yet to be identified. Besides repressing the m⁶A deposition in internal exons, pre-mRNA splicing also inhibits the m⁶A deposition in the last exons according to our deep learning modeling (**Fig. 8e**). However, EJC depletion did not affect m⁶A deposition at last exons (**Fig. 8f, and Supplementary Fig. 19c**). In addition, the loss of EJC could only increase the m⁶A signal on a small proportion of short internal exons (**Fig. 9**). Altogether, EJC, as a trans-factor, only contributes to m⁶A inhibition in a small proportion of short internal exons, suggesting that additional factors which may also participate in m⁶A deposition site-specificity are yet to be identified. We have added this new result as Fig. 8 and Fig. 9 to the manuscript thanks to the reviewer's insightful suggestion and encouragement.

5. Can the authors comment on whether the majority of "exon heterogeneity" is mostly attributed to DRACH density, or if there are additional factors? For example, the authors can characterize the exon lengths of cluster 1 and cluster 2 (Fig 4), which are known to be a large contributing factor to m6A deposition. Additionally, can the authors comment on

the 3' UTR length, which is thought to be positively associated with m6A?

We thank the reviewer for the suggestions. To systematically perform *de novo* analysis of the *cis*-elements, we examined all the pentamer occurrence comparing Cluster1 vs. Cluster2 of last exons. The NRACN motifs were more likely to be enriched in C1 exons (**Fig. 4m, and Supplementary Fig. 8m**). Additionally, m⁶A enhancers were also more enriched in C1 exons, while the m⁶A silencers were more avoided (**Fig. 4n, and Supplementary Fig. 8n**), supporting our finding that C1 exons tend to be in a better local *cis*-element environment than C2 exons. We also identified the 20 most enriched or avoided motifs. The 20 most enriched motifs included mostly part of the RRACH motif (**Fig. 4o, and Supplementary Fig. 8o**), and the 20 most avoided motif contained many CG dinucleotides (**Fig. 4p, and Supplementary Fig. 8p**). We also compared the exon lengths and 3'-UTR lengths between C1 and C2 last exons. Both exon length and 3'-UTR length of C1 last exons were longer than C2 last exons (**Response Fig. 2**). We have included this figure as **Supplementary Fig. 10**.

Response Figure 2: Density plot of exon length (Response Fig. 2a for mouse, and Response Fig. 2c for human), and 3'-UTR length (Response Fig. 2b for mouse, and Response Fig. 2d for human) of last exons. The density was compared between Cluster1 and Cluster2. The P-values were calculated by the Kolmogorov–Smirnov test (Significance: *** $p < 0.001$). The median values were labeled.

6. The authors show that latent m6A sites are more conserved than no-change sites. These latent m6A sites are often near splice sites. It is possible that these splice sites are conserved due to splicing, rather than m6A-related biology. Can the authors show a side-by-side comparison of the conservation of all splice sites to see how much the m6A machinery contributes to the conservation compared to the contribution of the splicing machinery?

We thank the reviewer for the advice. To compare the conservation between latent m⁶A sites and no-change sites, we also split the sites into two groups: near splice site (≤ 100 nt), and away from splice site (> 100 nt). We found that latent m⁶A sites were more conserved than no-change sites in both of two groups, regardless of their distance to splice sites (**Fig. 1e,f,k,l, Fig. 2l,m, Supplementary Fig. 1e,f,k,l, and Supplementary Fig. 4l,m**). These results demonstrated that m⁶A functionality contributes to the observed conservation of m⁶A sites instead of the contribution of splicing machinery.

7. The authors suggest that C1 exons need to be hidden from m6A as they encode a particular set of codons, many of which look like the “RAC” motif (amino acids D, N, and T). I wonder which came first evolutionarily. Did the splice site evolve, therefore blocking methylation at these sites, allowing more RAC motifs/codons to appear, or did these methylation sites evolve first, requiring splice sites to cover to prevent m6A deposition and therefore mRNA degradation?

We appreciate the reviewer’s deep thoughts into this question. As it is a very fascinating question in natural evolution, we are not sure the time order and both scenarios could be true. To be comprehensive, we added more writing to the manuscript to cover both situations so that readers could have a balanced view of both possibilities.

8. To me, one of the most fascinating aspects of the study, which also distinguishes it from the other competing published papers, is that this manuscript is very clear that only a subset of sites are influenced by splicing. It would be great to document this even more. Some -simple- experiments could include showing some individual examples of splicing junctions where there is an adjacent DRACH (GGACU) sequence, yet it is highly

methyated. A broader computational experiment could involve calculating the number of DRACH sites within 10 nt or so of an exon junction that have “above-median” m6A stoichiometry (or a similar analysis). This can really document the prevalence of DRACH sites that escape. One thing that I would suggest that the authors consider is whether the manuscript could be re-pitched to show that DRACH sites near exon junctions are in fact, NOT, always suppressed. Since there are three competing manuscripts, the authors may consider the unique strengths of this manuscript, which highlights that the EJC suppressor model does not fully explain m6A stoichiometry. To me, showing that exon junction complexes are modulatory, but not fully determinative, is a very important advance in the field. Related to this, can the authors speculate why some exon junctions are repressors for m6A deposition while others are not? This will be asked by most readers of this paper.

We again greatly appreciate the reviewer for the very helpful suggestions and insights to improve our manuscript. To echo the reviewer, we added two new figures (**Fig. 8 and 9**) to the manuscript by re-analyzing the m⁶A-IP in control or EJC depletion HEK293 cells (Uzonyi et al, 2023, Mol Cell, DOI:10.1016/j.molcel.2022.12.026). For expressed genes in HEK293 cells, about 0.4% short internal exons (≤ 200 nt) had m⁶A modification (280 out of 73,456 expressed short internal exons) (**Fig. 9a**) in control HEK293 cells, highlighting that there are m⁶A sites in these short exons that escaped pre-mRNA splicing inhibition. Upon the Y14 EJC component depletion, methylated short internal exons increased to 14.3% (**Fig. 9b**), in contrast to the fact that most of short exons were not subjected to EJC inhibition (the actual proportion of short internal exons that have RAC sites is as large as 94.5%) (**Fig. 9c**). These findings supported that EJC only contributed to m⁶A deposition inhibition in a small subset of short internal exons. This result highlighted that there would be other trans-factors that may contribute to m⁶A deposition inhibition yet to be discovered, and also as we showed throughout the manuscript, local cis-element environment (i.e. high m⁶A enhancer and low m⁶A silencer) may be also very important for the selective inhibition observed.

It's a very interesting question why some exon junctions are repressors for m⁶A deposition while others are not. Our current work reveals that there is a high degree of exon

heterogeneity for the m⁶A deposition inhibition by pre-mRNA splicing. We found that exons that are subjected to strong inhibition (i.e. C1 exons) tend to have good local *cis*-element composition, e.g. higher density of RAC motifs (**Fig. 4k-m, and Fig. 5k-m**), higher density of m⁶A enhancers (**Fig. 4n, and Fig. 5n**) and lower density of m⁶A silencers (**Fig. 4n, and Fig. 5n**). All these findings suggests that EJC may only play a modulatory rule in partially contributing to m⁶A site-specificity, other factors including local *cis*-element composition and beyond are yet to be characterized.

9. The authors argue that internal exons methylation and terminal exon methylation is similar. However, I believe this is not correct. The long internal exons have high methylation throughout the internal region of the long exon. However, the terminal exon only has methylation ~100-200 nt from the beginning of the exon, and then lack methylation throughout the remainder of the terminal exon (which the authors previously showed is typically long). Doesn't this demonstrate very different properties and mechanisms for terminal exons?

We apologized for the possible writing confusion here. The sentence that we wrote “internal exons and terminal exons share similar methylation mechanism” only means that the two share a similar rule of the local *cis*-element composition (please see Fig. 2g in Luo et al, 2022, Nat Comms, DOI: 10.1038/s41467-022-30209-7). Based on this analysis of local *cis*-elements regulating m⁶A site-specificity, we wrote that last exons and internal exons follow the same local *cis*-element rule in governing m⁶A deposition. To avoid confusion to readers, we re-wrote this part in the manuscript to clarify this part. The m⁶A site-specific methylation is primary determined by the flanking nucleotide sequence, and local functional *cis*-elements mainly resides within the 50 nt downstream of the site (Luo et al, 2022, Nat Comms, DOI: 10.1038/s41467-022-30209-7).

Indeed, the regional distribution of m⁶As in last exons and long internal exons are different (Ke et al, 2015, Genes Dev, DOI: 10.1101/gad.269415.115; Ke et al, 2017, Genes Dev, DOI: 10.1101/gad.301036.117). For internal exons, m⁶As are enriched in throughout the internal region of the long exon. For last exons, m⁶As are enriched when entering last exons. iM6A

deep learning modeling recapitulated the global distribution pattern of the m⁶A sites in mRNA (**Fig. 1a**), but the precise mechanism of this regional distribution is yet to be established, and we are working on this question as one of the future directions.

10. For many readers the im6A deep learning method is not clear, and it is not clear how the authors can remove introns/exons and model different patterns of methylation.

Schematics and figures would be helpful, perhaps to start the manuscript.

We thank the reviewer for the advice. In the wild-type situation, the input for iM6A is the nucleotide sequence of pre-mRNA, and the output is the probability of each nucleotide being a m⁶A site. In the intron deletion modeling situation, the input for iM6A is the nucleotide sequence of mRNAs (only exon sequences with no introns, exon/intron information is based on genomic annotation). To provide a clear description of the data analysis, we have provided more details to the schematics of **Fig. 1a**, and **Fig. 2a**, and the text are modified accordingly.

11. I don't believe that the one wet experiment supports the model - note that in Fig 3C, A1 goes up after intron removal and A4 does not. This does not make sense. In Fig 3D, shouldn't A3 go up with exon removal? Perhaps the SELECT assay is noisy, but the results were not completely helpful. Comments on the discrepancies and explanations are needed.

We thank the reviewer for the advice. We first modeled the m⁶A deposition in minigenes by iM6A modeling (**Supplementary Fig. 7a**), and then validated the predictions by SELECT assay (**Fig. 3c,d**). For *Lrp12* minigene (**Fig. 3c**), methylation of A1 was predicted to be increased after intron removal, while A4 was not (**Left panel of Supplementary Fig. 7a**). And the experimental validation matched the prediction (**Fig. 3c**). For *Gne* minigene, experimental validation showed that A4, and A5 had increased methylation after intron removal (**Fig. 3d**), which were prediction by iM6A modeling (**Right panel of Supplementary Fig. 7a**). Overall, eight RAC sites were predicted to increase their m⁶A level when the minigene did not undergo pre-mRNA splicing to produce the mRNA with the same nucleotide sequence (**Supplementary Fig. 7a**), and five such RAC sites were experimentally confirmed to increase

their m⁶A level (**highlighted in Fig. 3c,d**). These experimental validations confirmed that intron splicing inhibits m⁶A deposition at adjacent exons.

Minor concerns and suggestions

1. In Fig 1 (and Fig 2), it is ambiguous what authors mean by “last exon start”, as the “last exon start” is different before and after mutation. They should use “original last exon start” or some other naming convention.

We thank the reviewer for the suggestion, and modified the text in manuscript and metaplot.

2. The authors should be more clear about what “the gene” (line 100) refers to. Are they performing a transcriptome-wide *in silico* mutation of every last intron, or on a specific gene?

Sorry for the misunderstanding. We performed a transcriptome-wide *in silico* mutation for every last intron. We have modified the text to make it more clear.

3. The schematic of the gene in Fig 2a makes the pink gene appear to have all introns except the last two introns.

Thanks for the advice, we have modified the schematic to show how we remove all introns in

Fig. 2a

4. The authors should mention the original mean intron length as a comparison to the truncated intron length of 400 nt (line 170)

Thanks for the suggestion. The mean intron length is ~4.8 kb or ~6 kb in mouse or human transcriptome respectively. We have added it to the manuscript.

5. Line 208 has a typo: “experimentally” instead of “experientially”

We thank for pointing out the typo, and have modified it in manuscript.

6. Fig 4a, d, g and j are missing Y-axis labels, and labels for the colorbar. Figs 4b, c, e, f, h, i, k and l should also be labelled with “cluster 1” or “cluster 2” to improve readability.

Thanks for the suggestions. The labels were added in figures. And Cluster1 and Cluster2 were also labeled in figures.

7. The correlations in fig 6b, c, d and e are not very apparent due to the scale of changes.

Plotting the log-scale half-life may show these correlations more clearly.

We thank for the suggestion and plot the scatter plots with $\log_2(\text{mRNA half-life})$ as the x-axis. It indeed showed much clearer correlation.

8. I think that some people are going to wonder why the initial studies by Ke seeking to identify adjacent sequences that influence m6A deposition did not come up with exon junction regions/sequences/motifs as de-enriched. It might be interesting to talk about why this did not come up in earlier work by Ke, and possibly others. It is possible that it was actually seen by other groups or by Ke, but they did not focus on it. If so, it would be interesting to mention if it is present in other people's data sets.

We thank the reviewer for the suggestion. We didn't find that splice site sequences are de-enriched in our previous work when focusing on adjacent sequence that regulates m⁶A deposition (Luo et al, 2022, Nat Comms, DOI: 10.1038/s41467-022-30209-7). As we focused on local regulation in this previous work, it might be the reason that we missed this important signal. We have no knowledge if any other group saw the de-enrichment of splice sites in m⁶A region. On the other side, in our previous work characterizing the m⁶A distribution around splice site (Ke et al, 2017, Genes Dev, DOI: 10.1101/gad.301036.117), we did find that m⁶A is de-enriched near splice site region (Figure 4A-4B in Ke et al, 2017, Genes Dev, DOI: 10.1101/gad.301036.117).

9. It may be worth citing the recent Murukami review in Molecular Cell that argued that "gene architecture" would regulate m6A deposition based on their analysis of the commonality of m6A sites in different maps, as well as the relationship between m6A and splice junctions. Do the authors' data concur with their argument that m6A maps between tissues should be very similar and that m6A is "hard-wired" by gene architecture? Does the authors' data support the idea that m6A is unlikely to be dynamic? It would be useful

to touch on these issues in the Discussion.

We thank the reviewer to remind us the Murukami review. We appreciate the “gene architecture” thoughts there and have included it in our manuscript discussion. Our deep learning modeling approach highlights that the m⁶A deposition site-specificity is overwhelmingly determined by primary nucleotide sequences which includes both local *cis*-element motifs but also long-range *cis*-element regulation such as pre-mRNA splicing. All these facts support the view that m⁶A is “hard-wired” in the genome by genomic sequences which echoes the Murukami review. Given that, the dynamic regulation of m⁶A might not be a phenomenon that could be observed in most m⁶As. It is analogous to the situation of pre-mRNA splicing that most of pre-mRNA splicing is constitutive splicing though there does exist alternative splicing as a minor group. There might be m⁶A dynamics, as it is hard to rule out this possibility completely; if so, it would be likely to exist in a relatively fewer number compared to the static m⁶A methylation, though the underlying functional importance is yet to be established. In the same vein, alternative splicing regulation is an important layer of tissue-specific gene expression, though its number is much fewer than that of constitutive splicing. As a young field of m⁶A RNA biology, these directions are all exciting future questions of great importance.

Reviewer #2 (Remarks to the Author):

In this manuscript, the authors utilized the deep learning method to model how splicing influences m⁶A deposition and revealed the pre-mRNA splicing inhibits m⁶A deposition. This explains why m⁶A is enriched in last exons and long internal exons. Experiments were well designed to confirm the findings of modeling, and m⁶A sites in minigenes were examined with single nucleotide resolution. In general, the manuscript is well-written, and shows broad interest for the field of m⁶A.

We greatly thank Reviewer #2 for very valuable comments and helpful suggestions. We have carefully revised the manuscript according to the comments which improved our manuscript. Below is our point-to-point response. The reviewer’s comments are in blue, our responses are in black.

1. For last intron deletion, only 4.3% of sites in the last exons showed increased m⁶A deposition. Overall, the sites in the distal region did not have m⁶A deposition changes. This makes the inhibitory roles of last intron unreal. To be more specific, the percentage can be calculated for the sites close to last exon start.

Thanks for the suggestion. The RAC sites in last exon showing increased m⁶A deposition by last intron deletion were enriched within ~100 nt region to last exon start, and only few sites were located in > 200 nt region to last exon start (**Fig. 1b, and Supplementary Fig. 1b**). To be specific, we counted the sites in the 500 nt region of last exon start in the positional plot, and only a proportion of RAC sites (~12% for mouse, ~14% for human) had an increase in m⁶A deposition by last intron deletion.

2. Due to the fact that m⁶A is enriched in long exons, do the last exons with different lengths show the same m⁶A deposition inhibition by the last intron?

We thank for the suggestion, and Reviewer #1 asked a related question. We split the exons into three groups based on length (≤ 200 nt, 200-400 nt, and ≥ 400 nt). For last exons, the increased m⁶A deposition of sites were enriched in the 100 nt region to last exon start for all three groups (**Supplementary Fig. 2**), showing m⁶A deposition inhibition by pre-mRNA splicing occurs near the splicing sites for both short and long exons.

3. In Figure 2, do the long internal exons show the same m⁶A deposition inhibition by intron splicing as short internal exons?

Thanks for the advice. For short internal exons (≤ 200 nt), the increase of m⁶A deposition by introns deletion appeared in the 100 nt region of exon ends (**Fig. 2d,e, and Supplementary Fig. 4d,e**). For long internal exons (200-400 nt, and ≥ 400 nt), the increased sites were also enriched in the 100 nt region of exon ends (**Fig. 2f-i, and Supplementary Fig. 4f-i**). To be comprehensive, we split up exons for all such analysis (**Fig. 2, and Supplementary Fig. 2, 3, 4, 9, 11, 15**). These results demonstrated that long internal exons show the same m⁶A deposition inhibition by intron splicing as short internal exons.

4. In the legends of Figure 2, a typo occurred, and the panels were not correctly labeled.

We thank for pointing the mistakes. We have carefully revised the manuscript and modified the text.

Reviewer #3 (Remarks to the Author):

In this work Luo et al investigate the relation between RNA splicing and m6A deposition. In their intro they lay out some of the disagreement in the field around this interesting topic of research (p.2 line 47-57). Several works noted the association between splicing and m6A with m6A appearing to lack in internal exons but not in the last exon or long internal exons. While some labs claimed that m6A may be affecting/regulating splicing, the authors propose a different hypothesis by which splicing inhibits m6A. In this work the authors lay out several claims to support this hypothesis.

Most of the work presented is based on computational analysis using the authors previously published iM6A algorithm (Luo et al Nat Comm 2022). iM6A is a CNN based DL model that using a window of genomic sequence to classify a location as an m6a site. In the original publication the authors performed saturation mutagenesis analysis to find positions downstream of the site were important for the classification. Here they instead replaced the upstream sequence introns with exons and find a large effect. They then go on to extract various statistics based on that, before turning to experimental validation. The validation involved a mini-gene reporter assay and two constructs - one for Lrp12 and another with Gne. The authors compute statistics over the identified sites in these areas and show some of them change between constructs that have the introns and those where the introns were removed.

Overall, we find the topic very interesting and we appreciate the work that went into this project. However, we have major concerns with the results and associated claims, as presented.

One fundamental issue we found is that the main conclusions are well supported by the data/analysis. The association between splicing and m6A deposition is definitely intriguing, as

is the suggestion that the causal relationship is splicing affects m6A deposition. However, the authors ignore the fact that a common factor/mechanism may affect both rather than one affects the other. Regardless, we are quite suspicious of the computational analysis as performed here. The model used here is the same as their previous publication but previously they only identified a strong effect of downstream sequence. This fact is somewhat glossed over. Why is that? One main reason is likely the analysis procedure - single nucleotide variation (original paper) vs. replacing whole regions (current manuscript). However, this kind of analysis is highly problematic: The authors completely ignore the fact that the model they use is a discriminative model trained on genomic sequence. By introducing large changes of sequences (whole exons/introns) they present to the model out of distribution sequences for which results may be completely wrong. Think of showing an image classifier with cats and dogs and then asking it to classify an image of a raccoon. They did not train on such sequences/raccoons. Thus, this reviewer remains highly skeptical about any downstream analysis presented here as "proof" while all we see are associations based on out of distribution sequences.

Related to that is the lack of sufficient experimental evidence. First, the experiments are done on only two sequences, which is hardly a proof for a global phenomena. Second, the results even on those sequences are not very convincing. Only a few sites change significantly and there is no clear pattern for those that change/not. Last but not least, the experimental design seems flawed: At the very least the authors should have tested a construct with the same/similar sequences genomic sequences w/wo the splicing occurring (e.g. mutating the splice sites).

In summary, we found the claim interesting, the topic important and we appreciate the amount of work that went into this manuscript. However, as a computational paper there is no novelty (published model, just a different analysis using it) and as a manuscript based on novel results, we don't believe the conclusions are well supported by the data/analysis.

We thank Reviewer #3 for valuable comments and the reviewer's recommendation that the work is interesting and important. We have carefully revised the manuscript according to the comments which improved our manuscript. Below is our response in details.

In our previous work (Luo et al, 2022, Nat Comms, DOI: 10.1038/s41467-022-30209-7), we trained iM6A, a deep learning model that models m⁶A deposition in pre-mRNA with high accuracy in single nucleotide resolution. However, as a common problem in AI field, it's difficult to interpret the deep learning black box for biological insights. i.e. in AI machine learning, the deep learning black box has learned all the rules governing the m⁶A site-specificity as it demonstrates to us its high accuracy (Here one needs to emphasize that deep learning has **learned** the rules instead of **memorized** the training data sets, thus deep learning black box is able to do out-of-distribution prediction as in all AI applications such as protein structure prediction that AlphaFold2 can predict protein structure by taking input of protein sequence that it has never seen before). However, the deep learning black box keeps these rules to itself as AI machine knowledge which is hardly understandable by human at a glance. Thus how to creatively understand these AI machine knowledge as we human knowledge (i.e. scientific molecular mechanism) is a cutting edge field in AI research, and the scientific contribution of our work hits right at this challenge by performing innovative *in silico* genetic mutations as systematic input perturbations to the AI black box, and systematically study the output of the AI black box as *in silico* experiments to discover scientific molecular mechanisms with unprecedentedly fast-pace and cost efficiency. In our initial study (Luo et al, 2022, Nat Comms, DOI: 10.1038/s41467-022-30209-7), we focused on the study of local *cis*-element regulation by implementing high-throughput *in silico* saturated mutational modeling and found that the *cis*-elements regulating the m⁶A deposition preferentially reside within the 50 nt downstream of the m⁶A sites. This novel finding was validated by both independent experiments as well as evolutionary conservation, attesting our strategy in molecular mechanism discovery via deep learning modeling. In this new study, we focused on long-range *cis*-element regulation by implementing innovative *in silico* intron removal genomic modeling and found that pre-mRNA splicing inhibits the m⁶A deposition near splice sites. Again, our finding was validated by our own experiments as well as experiments from three

different labs independently which were very recently published online during our manuscript review (see details in the next paragraphs as well as the reviewer 1 comments). Our studies represent major methodology advances for novel computational strategies in interpreting deep learning black box and establish a fast-paced and cost-effective regulatory mechanism discovery method via *in silico* deep learning modeling.

Our finding was verified by experiments. We first modeled the m⁶A deposition in minigenes by iM6A modeling (**Supplementary Fig. 7a**), and then validated the predictions by SELECT assay (**Fig. 3c,d**). These single nucleotide resolution experimental validations confirmed that intron splicing inhibits m⁶A deposition at adjacent exons. During the review period of our manuscript, three independent papers published online found that exon junction complex (EJC) could contribute to the pre-mRNA splicing/exon-intron boundary inhibition of m⁶A (Yang et al, 2022, Nat Comms, DOI: 10.1038/s41467-022-35643-1; Uzonyi et al, 2023, Mol Cell, DOI: 10.1016/j.molcel.2022.12.026; He et al, 2023, Science, DOI: 10.1126/science.abj9090). We re-analyzed their m⁶A-IP in control or EJC depletion cells and found that their EJC depletion data could support our finding of m⁶A inhibition by pre-mRNA splicing/exon-intron boundary in short internal exons (**Fig. 8, Fig. 9, and Supplementary Fig. 19**), highlighting that the effectiveness of novel biological mechanism discovery via AI modeling. All these experimental work from three different labs in the field provides independent experimental support to our finding which is discovered in a completely new approach via *in silico* deep learning modeling.

In summary, our work represented a major conceptual advance for computational characterization of a long-range nucleotide sequence regulation mechanism of m⁶A-site specificity. It also provided a completely new mechanism that pre-mRNA splicing/exon-intron boundary influences mRNA stability through inhibiting nearby mRNA m⁶A modification.

REVIEWER COMMENTS

Reviewer #1 (Remarks to the Author):

I find this manuscript much improved and the authors have addressed the key concerns raised by me in the original submission.

I would like to make a comment that I think the authors should seriously consider and make textual revisions. In the original review, I wanted to understand how the authors came to the conclusion that pre-mRNA splicing was regulating m6A, when they only see a relationship between m6A and exon boundaries. They now add a sentence to explain this: (lines 85-87) "As pre-mRNA splicing is the only known mechanism that defines exon-intron boundary, we viewed these m6A sites being inhibited by pre-mRNA splicing or exon-intron boundary."

I think this is not correct. Here is a simple mechanism that does not involve splicing: binding of snRNAs and splicing machinery to splice junctions. These are the proteins that bind -before- splicing. These proteins could have an inhibitory effect on m6A rather than the splicing itself. Another example: H3K36me3 is a histone mark that is enriched on exons. As the polymerase encounters an exon, it encounters this mark, and the encounter may have an inhibitory effect.

Therefore I think it is safer to refer to exon boundaries only in the text, and then in the Discussion, the authors can refer to pre-mRNA splicing as a very plausible mechanism by which the boundaries can influence m6A deposition, but other possibilities could be involved.

This would involve changing pre-mRNA splicing to exon boundaries throughout the Results section.

As I said before, the authors previously showed that m6A deposition occurs -before- splicing, so the mechanisms I mentioned could be plausible if the authors previous findings are correct.

Minor

1.Line 485 and 488 - the authors are referring to "Sides". Is this the 5' or 3' side of the exon? Or is this like saying "on one hand...on the other hand..." if so, please switch to "hand" or preferably not use figurative language at all.

2.Many paragraphs are over a page long - readability would be improved if each paragraph was shorter with one idea.

3. The authors refer to papers with the author's first name (e.g. "Shino Murukami review" - they should refer to author's last names and provide citations.

4. "It's" should be "It it"

Reviewer #2 (Remarks to the Author):

Authors have addressed all my early concerns. I believe it is interesting work for the community of RNA modifications.

Reviewer #3 (Remarks to the Author):

In this revision Lou et al addressed many of the comments raised by myself and the two other reviewers. They did a good job updating the manuscript and I enjoyed reading it.

The main points to make regarding this revision are:

(1)

Reviewer #1 thoughtful comments and suggestions were a sheer enjoyment to read and ponder over. The authors did a good job addressing these suggestions with nice discussion points added and analysis of the recently published datasets. Together these made the manuscript both stronger (by showing data from these studies support the conclusions here) and also highlight what are the relative contributions of this work compared to recent ones (the model, the fact EJC do not explain all changes etc.).

(2)

There were three main points we raised previously:

(a)

There is no clear causal relationship proved by the DL model between splicing and m6A deposition: This point was raised by other reviewers as well and have been mostly adequately addressed by revising claims/statements and the addition of support by recent other works.

(b)

The DL model predictions can not be seen as a "proof" for a genome wide phenomena and should be taken with a grain of salt as the sequences introduced to it represent a strong deviation from the naturally occurring ones it was trained on: Here the authors strongly pushed back in their response to this critique so we will address this point even though in the grand scheme of things we don't think it matters much given the context of the recent papers that are now discussed/analyzed.

The authors state (with bold) the DL model "has learned the rules instead of memorized the training data sets..... [since it's tested on] sequence that it has never seen before".

This response represents, to us, a very limited understanding of DL models and the field in general. This

is, maybe, due to a misunderstanding of the term “out of distribution”. Let us elaborate here: The fact a DL is tested on a sequence it hasn’t seen before (the authors seem to quote basic ML courses here) is hardly a proof. The issue is these are discriminative not generative models and if you test them on input which is *very different* from what they have trained on (i.e. “out of distribution”) they may not perform as well. This is a well known problem in ML and an area of active research (see for example Shen et al 2021 <http://arxiv.org/abs/2108.13624>).

Putting aside theory, there are clear examples of recent significant fails in the structure prediction domain the authors cite (they refer to AlphaFold) specifically for RNA DL models where authors claimed high prediction power on unseen test sequences, while in practice they tested on unseen sequences with very similar structural features (e.g. Flamm et al 2022 <https://www.frontiersin.org/articles/10.3389/fbinf.2022.835422>)

The reason we emphasize this point is that we highly appreciate the effort and work made here (and previous work) by the authors. It is exactly for that reason that we think it is important that they too understand and acknowledge the limitations of their work/approach, and tone down claims/statements accordingly. Indeed, it seems the authors very much agree with the limitations of DL model based conclusions. As they themselves wrote nicely:

(Line 69):

“Then if the followed wet experiments validate randomly selected simulations, this contributes to verifying the model and the biological hypotheses it is designed to investigate”.

Which brings us to the next point:

(c)

Limited experimental support for the model’s conclusion:

This point was raised by other reviewers as well and unfortunately was mostly glossed over in the response. Also no new experiments were added. We hold that the experimental results are rather weak/limited and while they do provide support it’s hard to claim these alone “verifying the model” for a transcriptome wide effect. Nonetheless, the paper, model, and results, are now put in the context of other papers/results, which are nicely discussed. Thus, we believe that the paper is now able to both supply support for the model and distinguish itself (with the predictive DL model and the associated analysis) from the other works.

In summary, we think the authors have done a good job addressing many of the comments raised and elevated much of the major concerns we and others raised about the validity of the conclusions.

A few minor adjustments we recommend making are:

Consider revising the abstract - as is we find it does not read well and is not as inviting to the non m6A experts. This should help the authors appeal to a larger audience.

Line 173: "To validate this" - clearly we can agree an in-silico model prediction based solely on associations does not validate a biological model. Consider "To test this hypothesis....".

Line 516-517: "By iM6A deep learning modeling we uncovered" seems again like a strong assertion. Consider "Applying iM6A modeling suggested that...."

REVIEWER COMMENTS

Reviewer #1 (Remarks to the Author):

I find this manuscript much improved and the authors have addressed the key concerns raised by me in the original submission.

I would like to make a comment that I think the authors should seriously consider and make textual revisions. In the original review, I wanted to understand how the authors came to the conclusion that pre-mRNA splicing was regulating m6A, when they only see a relationship between m6A and exon boundaries. They now add a sentence to explain this: (lines 85-87) "As pre-mRNA splicing is the only known mechanism that defines exon-intron boundary, we viewed these m6A sites being inhibited by pre-mRNA splicing or exon-intron boundary."

I think this is not correct. Here is a simple mechanism that does not involve splicing: binding of snRNAs and splicing machinery to splice junctions. These are the proteins that bind -before- splicing. These proteins could have an inhibitory effect on m6A rather than the splicing itself. Another example: H3K36me3 is a histone mark that is enriched on exons. As the polymerase encounters an exon, it encounters this mark, and the encounter may have an inhibitory effect.

Therefore, I think it is safer to refer to exon boundaries only in the text, and then in the Discussion, the authors can refer to pre-mRNA splicing as a very plausible mechanism by which the boundaries can influence m6A deposition, but other possibilities could be involved. This would involve changing pre-mRNA splicing to exon boundaries throughout the Results section.

As I said before, the authors previously showed that m6A deposition occurs -before- splicing, so the mechanisms I mentioned could be plausible if the authors previous findings are correct.

We thank Reviewer #1's valuable comments which improved our manuscript. We appreciated the suggestion that exon-intron boundary is better in the text and have carefully revised the manuscript according to the comments.

Minor

1. Line 485 and 488 - the authors are referring to "Sides". Is this the 5' or 3' side of the exon? Or is this like saying "on one hand...on the other hand..." if so, please switch to "hand" or preferably not use figurative language at all.

Thanks for the suggestion. We have modified the text. "on one hand...on the other hand..." was used in the sentence.

2. Many paragraphs are over a page long - readability would be improved if each paragraph was shorter with one idea.

We thank for reviewer's suggestion. We have modified the manuscript accordingly.

3. The authors refer to papers with the author's first name (e.g. "Shino Murukami review" - they should refer to author's last names and provide citations.

Thanks for the suggestion. We have listed the authors' last names and provided citations.

4. "It's" should be "It is"

Thanks for the suggestion. We have corrected it accordingly.

Reviewer #2 (Remarks to the Author):

Authors have addressed all my early concerns. I believe it is interesting work for the community of RNA modifications.

We are thankful to Reviewer #2 for the recommendation of this manuscript for publication.

Reviewer #3 (Remarks to the Author):

In this revision Lou et al addressed many of the comments raised by myself and the two other reviewers. They did a good job updating the manuscript and I enjoyed reading it.

The main points to make regarding this revision are:

(1) Reviewer #1 thoughtful comments and suggestions were a sheer enjoyment to read and ponder over. The authors did a good job addressing these suggestions with nice discussion points added and analysis of the recently published datasets. Together these made the manuscript both stronger (by showing data from these studies support the conclusions here) and also highlight what are the relative contributions of this work compared to recent ones (the model, the fact EJC do not explain all changes etc.).

(2) There were three main points we raised previously:

(a) There is no clear causal relationship proved by the DL model between splicing and m6A deposition: This point was raised by other reviewers as well and have been mostly adequately addressed by revising claims/statements and the addition of support by recent other works.

(b) The DL model predictions can not be seen as a “proof” for a genome wide phenomena and should be taken with a grain of salt as the sequences introduced to it represent a strong deviation from the naturally occurring ones it was trained on: Here the authors strongly pushed back in their response to this critique so we will address this point even though in the grand scheme of things we don’t think it matters much given the context of the recent papers that are now discussed/analyzed.

The authors state (with bold) the DL model “has learned the rules instead of memorized the training data sets..... [since it’s tested on] sequence that it has never seen before”.

This response represents, to us, a very limited understanding of DL models and the field in general. This is, maybe, due to a misunderstanding of the term “out of distribution”. Let us elaborate here: The fact a DL is tested on a sequence it hasn’t seen before (the authors seem to quote basic ML courses here) is hardly a proof. The issue is these are discriminative not generative models and if you test them on input which is *very different* from what they have trained on (i.e. “out of distribution”) they may not perform as well. This is a well known

problem in ML and an area of active research (see for example Shen et al 2021 <http://arxiv.org/abs/2108.13624>).

Putting aside theory, there are clear examples of recent significant fails in the structure prediction domain the authors cite (they refer to AlphaFold) specifically for RNA DL models where authors claimed high prediction power on unseen test sequences, while in practice they tested on unseen sequences with very similar structural features (e.g. Flamm et al 2022 <https://www.frontiersin.org/articles/10.3389/fbinf.2022.835422>)

The reason we emphasize this point is that we highly appreciate the effort and work made here (and previous work) by the authors. It is exactly for that reason that we think it is important that they too understand and acknowledge the limitations of their work/approach, and tone down claims/statements accordingly. Indeed, it seems the authors very much agree with the limitations of DL model based conclusions. As they themselves wrote nicely: (Line 69): “Then if the followed wet experiments validate randomly selected simulations, this contributes to verifying the model and the biological hypotheses it is designed to investigate”.

Which brings us to the next point:

(c) Limited experimental support for the model’s conclusion: This point was raised by other reviewers as well and unfortunately was mostly glossed over in the response. Also no new experiments were added. We hold that the experimental results are rather weak/limited and while they do provide support it’s hard to claim these alone “verifying the model” for a transcriptome wide effect. Nonetheless, the paper, model, and results, are now put in the context of other papers/results, which are nicely discussed. Thus, we believe that the paper is now able to both supply support for the model and distinguish itself (with the predictive DL model and the associated analysis) from the other works.

In summary, we think the authors have done a good job addressing many of the comments raised and elevated much of the major concerns we and others raised about the validity of the conclusions.

We appreciate Reviewer #3's recognition of the revision and thank Reviewer #3 for the valuable comments and the support. We have carefully revised the manuscripts according to the comments by Reviewer #3.

A few minor adjustments we recommend making are:

Consider revising the abstract - as is we find it does not read well and is not as inviting to the non m6A experts. This should help the authors appeal to a larger audience.

We appreciate this suggestion and have added a few introductory sentences to the abstract.

Line 173: "To validate this" - clearly we can agree an in-silico model prediction based solely on associations does not validate a biological model. Consider "To test this hypothesis....".

Line 516-517: "By iM6A deep learning modeling we uncovered" seems again like a strong assertion. Consider "Applying iM6A modeling suggested that...."

We thank for both suggestions, and we have modified the text accordingly.